# Plasma biomarkers predict Alzheimer's disease before clinical onset in Chinese cohorts

Huimin Cai [1], Yana Pang[1], Xiaofeng Fu [1], Ziye Ren [1] & Longfei Jia [1]✉

Plasma amyloid-β (Aβ)42, phosphorylated tau (p-tau)181, and neurofilament light chain (NfL) are promising biomarkers of Alzheimer's disease (AD). However, whether these biomarkers can predict AD in Chinese populations is yet to be fully explored. We therefore tested the performance of these plasma biomarkers in 126 participants with preclinical AD and 123 controls with 8–10 years of follow-up from the China Cognition and Aging Study. Plasma Aβ42, p-tau181, and NfL were significantly correlated with cerebrospinal fluid counterparts and significantly altered in participants with preclinical AD. Combining plasma Aβ42, p-tau181, and NfL successfully discriminated preclinical AD from controls. These findings were validated in a replication cohort including 51 familial AD mutation carriers and 52 non-carriers from the Chinese Familial Alzheimer's Disease Network. Here we show that plasma Aβ42, p-tau181, and NfL may be useful for predicting AD 8 years before clinical onset in Chinese populations.

Alzheimer's disease (AD) is the predominant cause of dementia and creates an enormous social and economic burden[1,2]. Approximately 50 million people worldwide are affected by dementia, and this number is expected to exceed 80 million by 2030[3]. However, there is currently no effective therapy to prevent or attenuate AD progression[4]. Neuropathological changes in AD occur two decades before clinical symptoms arise, suggesting that early intervention may postpone the clinical presentation or development of AD[5]. Such early intervention requires early detection during the preclinical phase; thus, a symptom-agnostic diagnostic or predictive scheme should be addressed. Moreover, recent disease-modifying clinical trials of AD seem attainable[6], and an accurate method for risk stratification and subsequent personalized treatment is necessary. Therefore, it is imperative to shift from the traditional clinical symptom-based diagnostic algorithms to a biological definition of AD.

The National Institute on Aging-Alzheimer's Association (NIA-AA) proposed a research framework based on the amyloid, tau, and neurodegeneration (ATN) classification system, including amyloid-mediated pathology (A), tau-mediated pathology (T), and neurode-

generation (N)[7]. The ATN framework was initially designed for evaluating cerebrospinal fluid (CSF) and imaging (magnetic resonance and positron emission tomography [PET]) data[7]. Biomarker results from CSF and imaging studies demonstrate excellent discriminative performance and have improved our understanding of AD[8,9]. However, the perceived invasiveness of CSF sampling, as well as the high cost and the inadequate availability of imaging scans limits the use of these techniques for detecting pathological processes in the brain, particularly in primary care centers and large-scale clinical trial screenings[10]. As a result, there is a keen interest in identifying blood-based biomarkers that are readily available, cost-effective, and minimally invasive. Emerging plasma ATN candidate biomarkers include amyloid-β (Aβ)42, Aβ40, and Aβ42/Aβ40 for the A component; phosphorylated tau (p-tau) for the T component; and total tau (t-tau) and neurofilament light chain (NfL) for the N component[11]. Using ultrasensitive technologies[12,13], studies have demonstrated that these plasma biomarkers can precisely differentiate AD from cognitively intact participants or those with other neurodegenerative diseases[14–18]. Furthermore, these plasma biomarkers showed an association with

[1]Innovation Center for Neurological Disorders and Department of Neurology, Xuanwu Hospital, Capital Medical University, National Clinical Research Center for Geriatric Diseases, Beijing, China. ✉e-mail: longfei@mail.ccmu.edu.cn

future AD[19–21]. However, most of the studies were based on Western populations. As race and ethnicity might influence the assessment of these biomarkers[22–25], it is imperative to validate their performance across diverse populations including Chinese. While a recent study[26] from China revealed that a combination of plasma Aβ42, p-tau181, and clinical features exhibited excellent diagnostic performance for AD, there remains uncertainty regarding whether they can predict AD prognosis in Chinese populations.

In this study, we measure the plasma levels of Aβ40, Aβ42, p-tau181, t-tau, and NfL in two independent Chinese cohorts and assess the association between plasma biomarkers and their CSF counterparts. Further, we evaluate the efficiency of plasma biomarkers in predicting AD in Chinese populations before cognitive decline.

## Results

### Participant characteristics

Cohorts 1 and 2 are two sub-cohorts selected from the China Cognition and Aging Study (China COAST) and the Chinese Familial Alzheimer's Disease Network (CFAN). Participants with a clinical diagnosis of AD and with CSF p-tau181/Aβ42 > 0.14 as well as CSF Aβ42 < 500 pg/ml were confirmed to have AD. Tables 1 and 2 present the demographic and clinical characteristics of the two cohorts. In cohort 1, 126 Chinese participants with AD and 123 normal controls were included, with 125 (50.2%) women and a mean (SD) age of 69.5 (6.7) years. As expected, the AD group had a significantly higher frequency of apolipoprotein E (*APOE*) ε4 than did the control group. In cohort 1, participants who were cognitively intact at baseline but developed AD at follow-up were determined to have preclinical AD at baseline. Those who consistently remained cognitively intact at baseline and follow-up were determined to be normal controls. Participants with mild cognitive impairment (MCI) at baseline or follow-up were not included. In cohort 2, 51 Chinese familial AD (FAD) mutation carriers and 52 non-carriers were included, with 53 (51.5%) women and a mean (SD) age of 45.6 (4.9) years. Carriers with FAD mutations were determined to have preclinical AD as they were virtually destined to develop AD[27]. Non-carriers within the families were determined to be normal controls. The average estimated years to symptom onset (EYO) in the mutation carriers was 10.2 (1.2). No significant difference was observed in educational attainment between groups in either cohort. There was no significant difference in Mini-Mental State Examination (MMSE) scores between the groups in cohort 1 at baseline or in cohort 2. Participants with AD in cohort 1 were cognitively normal at baseline (preclinical AD) but cognitively declined at follow-up (MMSE changed from 29.1 to 20.7).

### Group differences between AD and control in plasma biomarkers

In cohort 1 at follow-up, plasma levels of Aβ42 and Aβ42/Aβ40 were significantly lower in the AD group (Aβ42: $P = 3.57 \times 10^{-38}$, $P_{BON} = 2.14 \times 10^{-37}$; Aβ42/Aβ40: $P = 1.42 \times 10^{-12}$, $P_{BON} = 8.52 \times 10^{-12}$; Fig. 1a, c) compared to the control groups, whereas plasma levels of p-tau181 and NfL were significantly higher in the AD group (p-tau181: $P = 1.04 \times 10^{-31}$, $P_{BON} = 6.24 \times 10^{-31}$; NfL: $P = 1.10 \times 10^{-14}$, $P_{BON} = 6.60 \times 10^{-14}$; Fig. 1e, f). Plasma Aβ40 and t-tau levels were not significantly different between the AD and control groups (Aβ40: $P = 0.02$, $P_{BON} = 0.12$; t-tau: $P = 0.71$, $P_{BON} > 0.99$; Fig. 1b, d).

### Correlations between plasma biomarkers and CSF counterparts

To validate whether plasma Aβ42, Aβ40, Aβ42/Aβ40, p-tau181, t-tau, and NfL levels can reflect CSF changes in AD, we performed a correlation analysis between plasma biomarkers and their CSF counterparts. Plasma levels of Aβ42, Aβ40, p-tau181, and NfL at follow-up were significantly correlated with their corresponding CSF levels in cohort 1 (Aβ42: adjusted $R^2 = 0.66$, $P = 3.15 \times 10^{-60}$; Aβ40: adjusted $R^2 = 0.41$, $P = 1.20 \times 10^{-30}$; p-tau181: adjusted $R^2 = 0.53$, $P = 2.86 \times 10^{-42}$; NfL: adjusted $R^2 = 0.58$, $P = 4.81 \times 10^{-49}$; Fig. 2a, b, d, e), whereas plasma t-tau showed a relatively weak association with CSF t-tau (adjusted $R^2 = 0.28$, $P = 2.21 \times 10^{-19}$; Fig. 2c).

### Predictive performance of plasma biomarkers for AD in cohort 1

At baseline, plasma levels of Aβ42 were significantly lower in participants with preclinical AD ($P = 1.88 \times 10^{-5}$, $P_{BON} = 1.13 \times 10^{-4}$; Fig. 3a), whereas plasma levels of p-tau181 and NfL were significantly higher in participants with preclinical AD than in controls in cohort 1 (p-tau181: $P = 3.38 \times 10^{-5}$, $P_{BON} = 2.03 \times 10^{-4}$; NfL: $P = 7.09 \times 10^{-6}$, $P_{BON} = 4.25 \times 10^{-5}$; Fig. 3e, f). Plasma Aβ40, Aβ42/Aβ40, and t-tau levels did not significantly differ between the two groups (Aβ40: $P = 0.21$, $P_{BON} > 0.99$; Aβ42/Aβ40: $P = 0.08$, $P_{BON} = 0.48$; t-tau: $P = 0.71$, $P_{BON} > 0.99$; Fig. 3b–d). Furthermore, the association between baseline and follow-up levels of plasma biomarker expression was evaluated. All biomarkers showed a significant association between baseline and follow-up levels (Aβ42: adjusted $R^2 = 0.69$, $P = 2.68 \times 10^{-64}$; Aβ40: adjusted $R^2 = 0.66$, $P = 1.17 \times 10^{-59}$; t-tau: adjusted $R^2 = 0.68$, $P = 9.03 \times 10^{-43}$; p-tau181: adjusted $R^2 = 0.62$, $P = 1.05 \times 10^{-53}$; NfL: adjusted $R^2 = 0.64$, $P = 2.16 \times 10^{-57}$; Fig. 4), indicating a concordant trend with future changes from baseline. To determine whether values of the plasma biomarkers were significantly changed within an individual between baseline and follow-up, we calculated Lin's concordance correlation coefficient (CCC). The CCC was 0.58–0.77, and the fitted lines of the

## Table 1 | Characteristics of Participants in Cohort 1[a]

| Characteristic | Total Sample (Follow-up, n = 249) | Baseline | | Follow-up | |
|---|---|---|---|---|---|
| | | Control (n = 123) | PreAD (n = 126) | Control (n = 123) | AD (n = 126) |
| Age, mean (SD), year | 69.5 (6.7) | 60.0 (7.1) | 59.0 (6.3) | 70.0 (7.1) | 69.0 (6.3) |
| Educational attainment, mean (SD), year | 9.1 (2.3) | 9.5 (2.2) | 8.8 (2.3) | 9.5 (2.2) | 8.8 (2.3) |
| Women, No. (%) | 125 (50.2) | 62 (50.4) | 63 (50.0) | 62 (50.4) | 63 (50.0) |
| *APOE ε4* status, No. (% positive) | 74 (29.7) | 22 (17.9) | 52 (41.3)[b] | 22 (17.9) | 52 (41.3)[b] |
| MMSE score, mean (SD) | 24.8 (4.6) | 29.0 (0.5) | 29.1 (0.5) | 29.0 (0.5) | 20.7 (2.8)[c] |
| CSF biomarkers, mean (SD), pg/ml | | | | | |
| Aβ42 | 545.9 (197.7) | / | / | 713.2 (135.1) | 382.6 (72.7)[d] |
| t-tau | 480.9 (209.4) | / | / | 327.1 (96.0) | 631.0 (179.0)[e] |
| p-tau181 | 89.7 (53.2) | / | / | 52.5 (25.5) | 123.2 (49.2)[f] |

*Aβ* amyloid-β, *AD* Alzheimer's disease, *APOE* apolipoprotein E, *CSF* cerebrospinal fluid, *MMSE* Mini-Mental State Examination, *PreAD* preclinical Alzheimer's disease, *p-tau* phosphorylated tau, *SD* standard deviation, *t-tau* total tau.
[a]Differences between the groups were measured using two-sided Student's t- and χ2 (sex and *APOE* ε4 status) tests without adjustment for multiple comparisons.
[b]$P = 5.40 \times 10^{-5}$, [c]$P = 7.26 \times 10^{-90}$, [d]$P = 7.29 \times 10^{-67}$, [e]$P = 3.41 \times 10^{-42}$, [f]$P = 8.16 \times 10^{-34}$ compared with control.

linear regression deviated from the 45° line. The means of each plasma biomarker were also significantly different between baseline and follow-up (Supplementary Table 1). These findings suggested that the biomarker levels significantly differed between baseline and follow-up.

A logistic model was constructed to investigate the performance of these plasma biomarkers in discriminating participants with AD from controls and those with preclinical AD from controls (Supplementary Table 2). After adjusting for age, sex, educational attainment, and *APOE* ε4 status, plasma Aβ42, p-tau181, and NfL were significant contributors to the occurrence of AD. Age, sex, and educational attainment were not statistically significant ($P > 0.05$) and were excluded from the final model. A multicollinearity examination among the

three plasma biomarkers revealed that all tolerances were >0.1, variance inflation factors (VIFs) were <10, eigenvalues were >0, and condition indices were <30, suggesting no significant multicollinearity existed among the three plasma biomarkers. The predictive values generated from the logistic model were further assessed using receiver operating characteristic (ROC) curve analysis. The areas under the curve (AUCs) of the three combined plasma biomarkers were 0.78 (95% confidence interval [CI]: 0.71-0.83, $P = 1.22 \times 10^{-13}$; Fig. 5) at baseline and 0.99 (95% CI: 0.98–1.00, $P = 4.20 \times 10^{-40}$; Fig. 5) at follow-up. The likelihood ratio test showed that adding *APOE* ε4 status to the models significantly improved the disease prediction accuracy at baseline but not at follow-up (Supplementary Table 2). The DeLong test further confirmed that adding *APOE* ε4 status significantly increased the AUC of the ROC analysis at baseline (AUC = 0.81, 95% CI: 0.75–0.86, $P = 5.11 \times 10^{-17}$, DeLong $P = 3.37 \times 10^{-2}$; Fig. 5 and Supplementary Table 2) but not at follow-up (AUC = 0.99, 95% CI: 0.98–1.00, $P = 1.89 \times 10^{-40}$, DeLong $P = 0.17$; Fig. 5 and Supplementary Table 2). Furthermore, the DeLong test showed that these models of combination were significantly superior to those of individual biomarkers (Supplementary Table 2). These results suggest that the model combining plasma Aβ42, p-tau181, and NfL displayed a good discriminative ability for both AD and preclinical AD.

### Replication in the familial AD cohort

To strengthen the validity of the findings, we used cross-sectional data selected from an FAD cohort (cohort 2), in which mutation carriers with an EYO of 8–10 years were determined to be participants with preclinical AD. The replication cohort replicated the main findings. Plasma levels of Aβ42 were significantly lower in mutation carriers than in non-carriers (Aβ42: $P = 9.45 \times 10^{-4}$, $P_{BON} = 5.67 \times 10^{-3}$; Fig. 6a), whereas plasma levels of p-tau181 and NfL were significantly higher in mutation carriers (p-tau181: $P = 1.74 \times 10^{-4}$, $P_{BON} = 1.04 \times 10^{-3}$; NfL: $P = 1.16 \times 10^{-3}$, $P_{BON} = 6.96 \times 10^{-3}$; Fig. 6e, f). Plasma levels of Aβ40,

**Table 2 | Characteristics of Participants in Cohort 2[a]**

| Characteristic | Total Sample (n = 103) | Control (n = 52) | Mut (n = 51) |
|---|---|---|---|
| Age, mean (SD), year | 45.6 (4.9) | 45.9 (5.5) | 45.2 (4.3) |
| Educational attainment, mean (SD), year | 9.2 (2.7) | 9.5 (3.1) | 8.9 (2.4) |
| Women, No. (%) | 53 (51.5) | 27 (51.9) | 26 (51.0) |
| MMSE score (SD) | 29.1 (0.6) | 29.0 (0.7) | 29.2 (0.6) |
| Estimated years to symptom onset, year | / | / | 10.2 (1.2) |
| Family mutation, n (%) | | | |
| *PSEN1* | / | / | 36 (70.6) |
| *PSEN2* | / | / | 4 (7.8) |
| *APP* | / | / | 11 (21.6) |

*APP* amyloid precursor protein, *MMSE* Mini-Mental State Examination, *Mut* mutation carrier, *PSEN* presenilin, *SD* standard deviation.

[a]Differences between the groups were measured using two-sided Student's *t*- and χ2 (sex and *APOE* ε4 status) tests without adjustment for multiple comparisons.

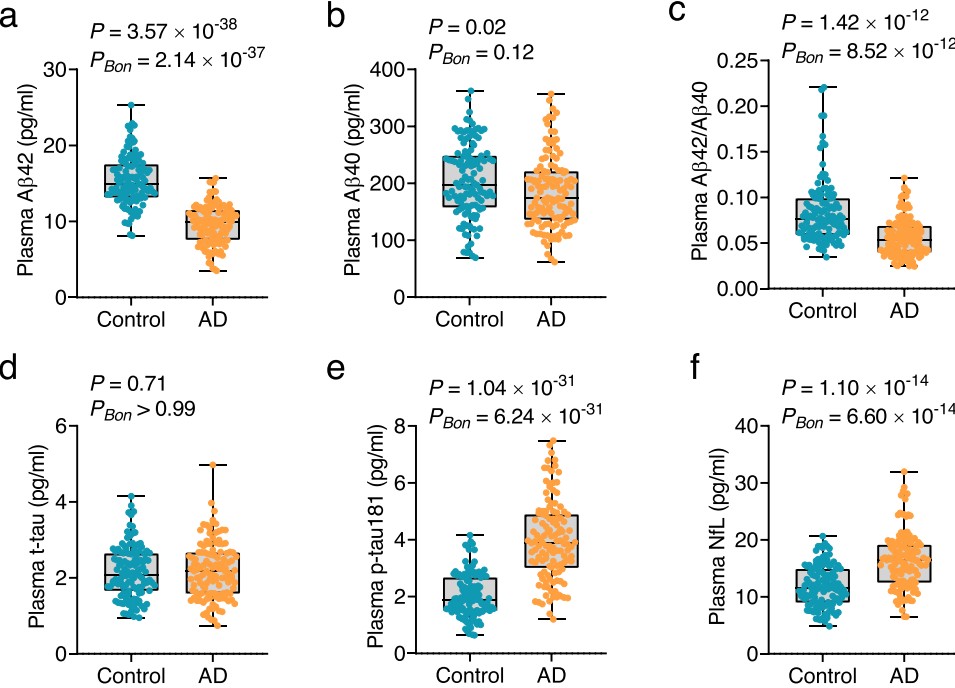

**Fig. 1 | Levels of plasma biomarkers in cohort 1 at follow-up.** Plasma levels of Aβ42 (**a**), Aβ40 (**b**), Aβ42/Aβ40 (**c**), t-tau (**d**), p-tau181 (**e**), and NfL (**f**) were measured in cohort 1 at follow-up. *P*-values and Bonferroni-corrected *P*-values ($P_{Bon}$) from two-sided Student's *t*-test comparing participants with AD (orange dots) and normal controls (blue dots) are shown separately in each panel. Box ends represent the 25th and 75th percentiles, and the horizontal line within each box indicates the median. Whiskers extend to the upper and lower adjacent values or the most extreme points within 1.5 × interquartile range of the 25th and 75th percentiles. n = 123 (control), 126 (AD). Aβ amyloid-β, AD Alzheimer's disease, Bon Bonferroni, NfL neurofilament light chain, p-tau phosphorylated tau, t-tau total tau. Source data are provided as a Source Data file.

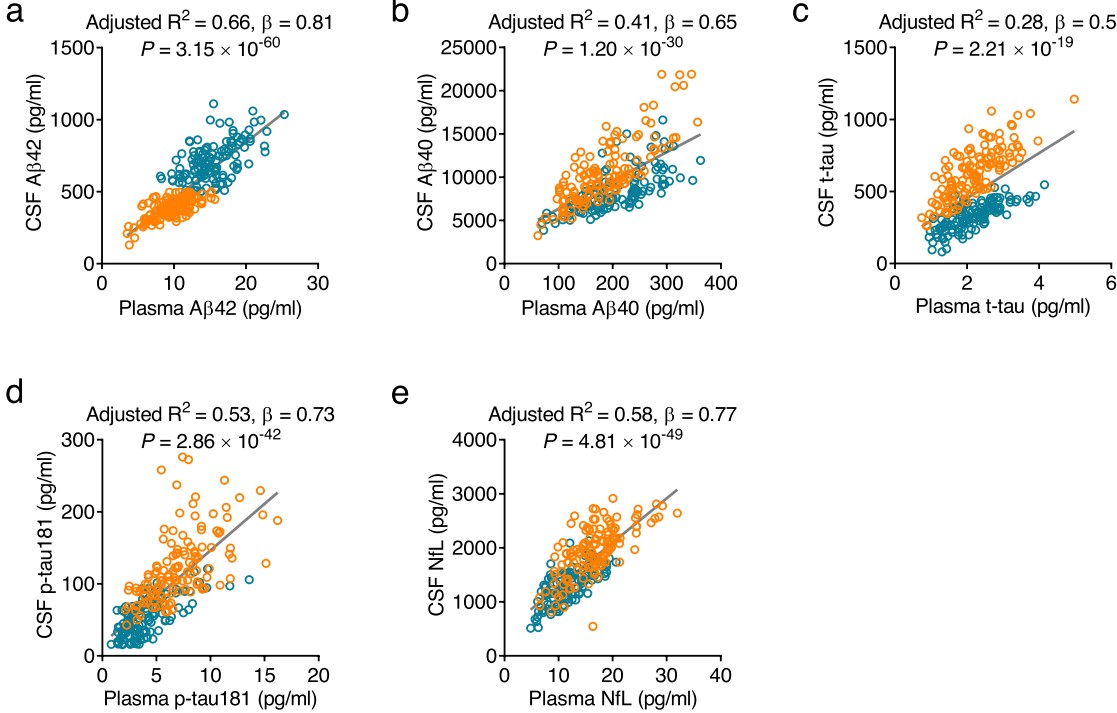

**Fig. 2 | Correlation between plasma biomarkers with cerebrospinal fluid (CSF) counterparts.** The concentrations of Aβ42 (**a**), Aβ40 (**b**), t-tau (**c**), p-tau181 (**d**), and NfL (**e**) in plasma are correlated with their counterparts in CSF in cohort 1 at follow-up. Adjusted $R^2$, β, and uncorrected $P$ values from two-sided linear regression are shown separately in each panel. Blue circles indicate controls while orange circles indicate participants with AD. Solid lines indicate the correlation. $n = 123$ (control), 126 (AD). Aβ amyloid-β, AD Alzheimer's disease, NfL neurofilament light chain, p-tau181 phosphorylated tau, t-tau total tau. Source data are provided as a Source Data file.

Aβ42/Aβ40, and t-tau did not significantly differ between the two groups (Aβ40: $P = 0.05$, $P_{BON} = 0.30$; Aβ42/Aβ40: $P = 0.78$, $P_{BON} > 0.99$; t-tau: $P = 0.94$, $P_{BON} > 0.99$; Fig. 6b–d). Moreover, the combination of plasma Aβ42, p-tau181, and NfL achieved an AUC of 0.79 (95% CI: 0.71-0.88, $P = 2.77 \times 10^{-7}$; Fig. 5) in discriminating mutation carriers from controls and significantly outperformed individual biomarkers (Supplementary Table 2), further supporting the predictive ability of these three plasma biomarkers for AD in Chinese populations. In addition, we separately performed ROC analyses in *APP*, *PSEN1*, or *PSEN2* mutation carriers and controls. The AUCs of each group were very similar, ranging between 0.78–0.80 (Supplementary Fig. 1).

## Discussion

This study examined two independent Chinese cohorts and explored the predictive capacity of plasma biomarkers in the preclinical phase of AD. We revealed that plasma Aβ42, Aβ40, p-tau181, and NfL levels were significantly correlated with their CSF counterparts in participants with AD and healthy controls, indicating that plasma concentrations of these biomarkers may reflect CSF changes. We also found that plasma Aβ42, p-tau181, and NfL levels were altered in the AD group and differentiated participants with AD from controls with high accuracy. Furthermore, the three plasma biomarkers were altered in participants with preclinical AD and, combined, could detect participants with preclinical AD at least 8 years before clinical onset. These findings indicate that plasma Aβ42, p-tau181, and NfL are reliable predictors of AD in Chinese populations.

To assess whether plasma biomarkers in the preclinical phase can predict the development of AD in Chinese populations, this study recruited participants from a longitudinal cohort who were cognitively intact 8–10 years before this study. Participants who developed AD with abnormal CSF levels of p-tau181/Aβ42 and Aβ42 were designated as preclinical AD. Simultaneous assessment of these biomarkers repeatedly collected from individual participants allows for analysis of these biomarkers to quantify their role in predicting AD development. Additionally, we validated the results by selecting mutation carriers from an FAD cohort with an EYO of 8–10 years (i.e., those who were destined to develop AD 8–10 years later). Based on the two independent cohorts, the final models including only a combination of plasma biomarkers yielded a consistent AUC of 0.78–0.79, and adding *APOE* ε4 status to the models only resulted in a slight increase in prediction accuracy, suggesting that the model was effective and can be generalized to both sporadic and familial AD in Chinese populations.

In line with our findings, several recent studies have shown that different combinations of plasma biomarkers, based on the ATN framework with basic demographics, can predict future AD 4.8–6 years later in cognitively intact participants[19,20]. A longitudinal study with 14 years of follow-up demonstrated that a combination of low Aβ42 and high NfL plasma levels was significantly correlated with AD risk[28]. In a community-based cohort with a 17-year follow-up assessment, a combination of plasma p-tau181, NfL, and glial fibrillary acidic protein achieved an AUC of 0.794 and 0.791 in the first 6 and 9 years, respectively, in predicting AD diagnosis; however, the AUC decreased to 0.712 between 9 and 17 years of follow-up[21]. Similarly, combining p-tau181, p-tau217, and Aβ42/Aβ40 predicted future conversion to AD dementia in patients with MCI[20]. Furthermore, with or without other measures, abnormal plasma biomarker levels exhibited high predictive accuracy for future AD-related endophenotypes, including worsening cognition, brain atrophy, and tau or amyloid burden evaluated by PET, in non-demented participants[19,29–32]. Strikingly, some plasma biomarkers are highly predictive of AD neuropathology, as confirmed by autopsy[33,34]. Although many studies have investigated the relationship between plasma biomarkers and AD, conducting research across ethnic and racial diversities is crucial, as these factors may influence the assessment of biological risks associated with AD[22–25]. This study was conducted among Chinese populations, which reinforces previous findings from European and American populations[19,20,35]. Our findings

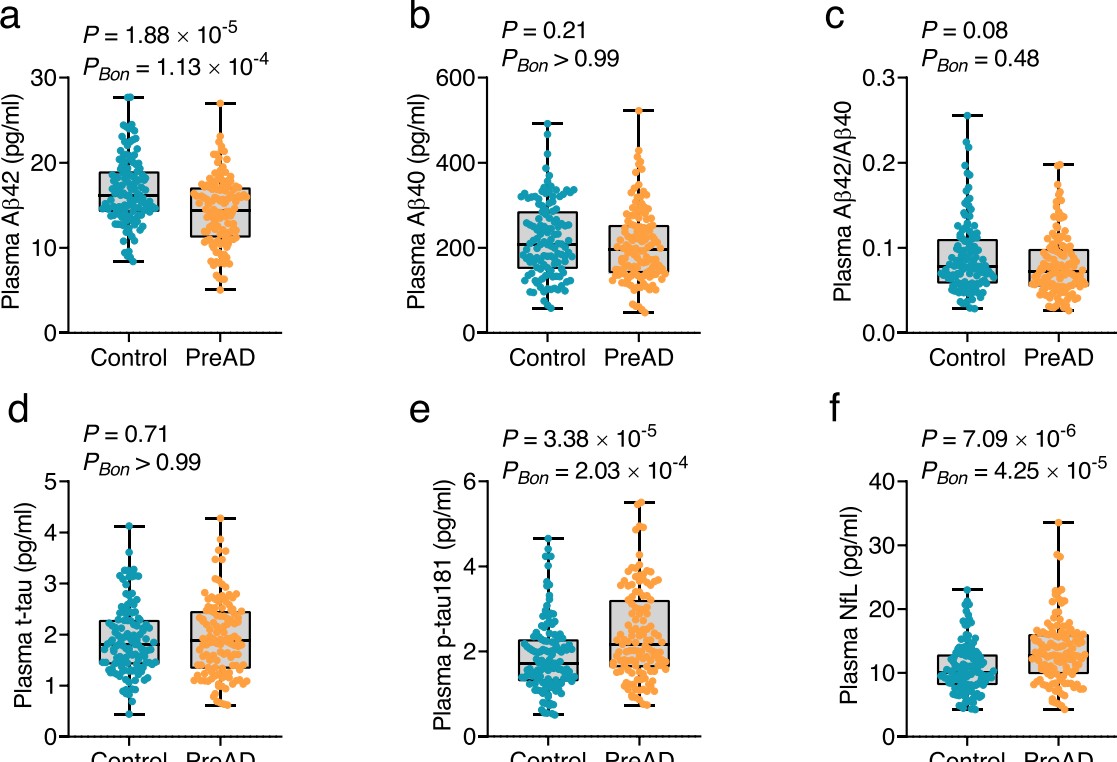

**Fig. 3 | Levels of plasma biomarkers in cohort 1 at baseline.** Plasma levels of Aβ42 (**a**), Aβ40 (**b**), Aβ42/Aβ40 (**c**), t-tau (**d**), p-tau181 (**e**), and NfL (**f**) were measured in cohort 1 at baseline. *P*-values and Bonferroni-corrected *P*-values ($P_{Bon}$) from two-sided Student's *t*-test comparing participants with PreAD (orange dots) and normal controls (blue dots) are shown separately in each panel. Box ends represent the 25th and 75th percentiles, and the horizontal line within each box indicates the median. Whiskers extend to the upper and lower adjacent values or the most extreme points within the 1.5 × interquartile range of the 25th and 75th percentiles. $n = 123$ (control), 126 (PreAD). Aβ amyloid-β, Bon Bonferroni, NfL neurofilament light chain, PreAD preclinical Alzheimer's disease, p-tau phosphorylated tau, t-tau total tau. Source data are provided as a Source Data file.

further add to the evidence of plasma biomarkers as reliable prognostic candidates for AD and highlight their promising roles in future clinical practice and research.

As expected, the predictive capacity of the combination of plasma biomarkers for AD was superior to that of the individual biomarkers. Some studies suggest that some of these biomarkers are easily affected by preanalytical procedures, and pairing them with other biomarkers might mitigate these confounds to some extent[36,37]. Furthermore, a simulated clinical trial demonstrated that a screening scheme using a combination of plasma biomarkers would lead to a greater reduction in sample size than using individual biomarkers[19]. Therefore, despite the ease of including only individual biomarkers, a combination of biomarkers is recommended for predicting AD.

As a hallmark of AD, plasma Aβ42/Aβ40 may be altered in the preclinical stage of AD. However, no significant difference was detected in the plasma levels of Aβ42/Aβ40 between participants with preclinical AD and controls in this study. Previous studies examining plasma Aβ42/Aβ40 in participants with preclinical AD have reported inconsistent findings. For example, some studies showed no significant change in plasma Aβ42/Aβ40 levels in participants with preclinical AD[38,39], which is similar to our study, while others showed that plasma Aβ42/Aβ40 levels were reduced in these individuals compared with controls[40,41]. This is an intriguing issue to be further explored in international multicenter studies. Moreover, plasma Aβ42/Aβ40 level did not contribute to the final predictive model. In line with our study, a cross-sectional study also found that plasma Aβ42/Aβ40 were not as efficient as plasma Aβ42 in differentiating participants with AD from those with MCI or controls[26]. This discrepancy might be attributed to different measurement platforms used across these studies, which is

supported by a head-to-head study[42] that found mass spectrometry-based methods outperformed immunoassays for evaluating plasma Aβ42/Aβ40 as an indicator of aberrant brain Aβ burden.

Aβ40 has been considered a stable normalizer during sample collection, processing, and analysis[43,44]. However, our study observed a non-significant downward trend in the plasma level of Aβ40 in participants with preclinical AD at baseline. Similar findings were previously reported in subjective cognitive decline, MCI, and Aβ-positive non-demented participants[20,26,41,45]. Studies have shown that different test methods can affect the results[42]; therefore, it is unclear whether the differences in results are due to distinct methods or even racial disparities across studies. The reason for the non-significantly lower levels of Aβ40 observed in our study and others may need further investigation in international multicenter studies.

Similarly, plasma t-tau was ineffective in discriminating preclinical AD in both group-level difference and predictive models, which is consistent with previous findings that plasma t-tau levels failed to show diagnostic value for AD[26] or an association with all-cause or AD dementia risk[28]. In contrast, evidence from the Framingham Heart Study has revealed that plasma t-tau increases the risk of incident AD dementia[32]. These contradictory findings could be explained by a large overlap in plasma t-tau levels between AD patients and the normal aging population[46], indicating that plasma t-tau might be a nonspecific biomarker for AD. Furthermore, we used the Innotest method for measurements of CSF t-tau, and Quanterix assays for plasma t-tau. The Innotest test for CSF t-tau is directed at mid-region epitopes, while the Quanterix Neurology 3-plex kit for plasma t-tau targets N-terminal-to-mid-region epitopes, which targets epitopes identical to the Quanterix Tau 2.0 kit used in the Framingham Heart Study[32]. Mass spectrometry

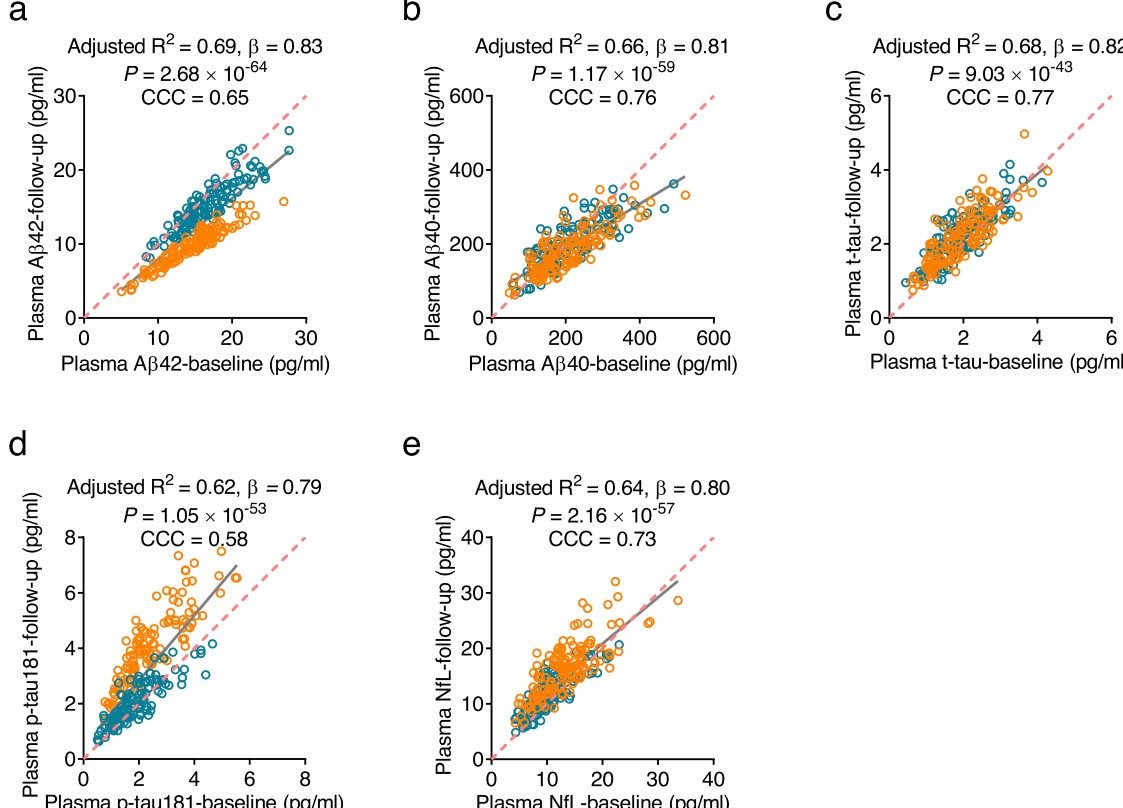

**Fig. 4 | Correlations between plasma biomarkers at baseline and follow-up levels in cohort 1.** The concentrations of plasma levels of Aβ42 (**a**), Aβ40 (**b**), t-tau (**c**), p-tau181 (**d**), and NfL (**e**) at baseline are significantly correlated with those at follow-up in cohort 1. Blue circles indicate controls while orange circles indicate participants with AD/PreAD. Adjusted $R^2$, β, and uncorrected $P$-values from two-sided linear regression are shown separately in each panel. Solid lines indicate the correlation. Dotted lines indicate the reference lines generated if the two measurements produce identical results and the CCCs are provided. $n = 123$ (control), 126 (AD/PreAD). Aβ amyloid-β, AD Alzheimer's disease, CCC concordance correlation coefficient, NfL neurofilament light chain, PreAD preclinical Alzheimer's disease, p-tau phosphorylated tau, t-tau total tau. Source data are provided as a Source Data file.

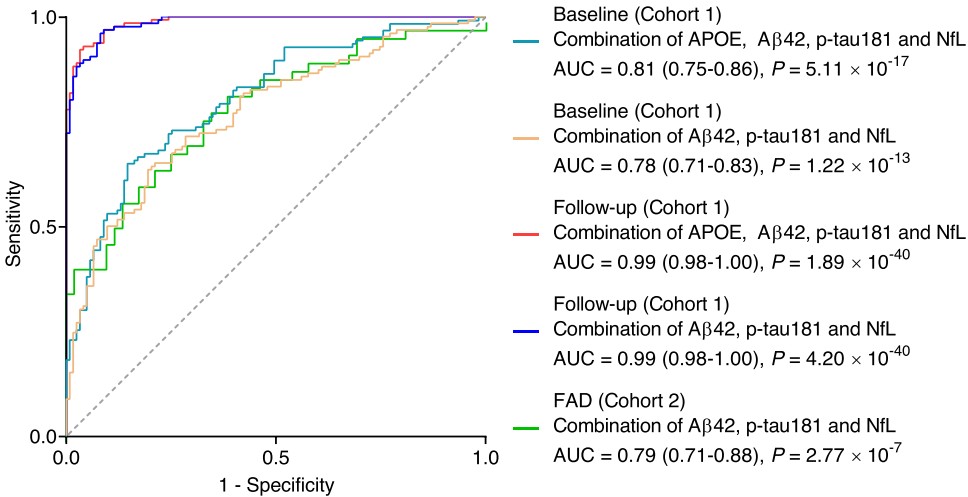

**Fig. 5 | Prediction of plasma Aβ42, p-tau181, and NfL for Alzheimer's disease (AD) in cohorts 1 and 2.** Receiver operating characteristic (ROC) curve analysis and corresponding areas under the curve (AUCs) from logistic regression models for plasma Aβ42, p-tau181, and NfL to assess accuracy when differentiating participants with AD from normal controls, PreAD from normal controls, and mutation carriers from normal controls. Models were generated from a combination of plasma Aβ42, p-tau181, and NfL with or without *APOE* ε4 status. All statistical analyses were two-sided without adjustment for multiple comparisons. $n = 123$ (control), 126 (AD/PreAD) in cohort 1; $n = 52$ (control), 51 (Mut) in cohort 2. Aβ amyloid-β, AD Alzheimer's disease, *APOE* apolipoprotein E, FAD familial Alzheimer's disease, NfL neurofilament light chain, Mut mutation carrier, PreAD preclinical Alzheimer's disease, p-tau phosphorylated tau. Source data are provided as a Source Data file.

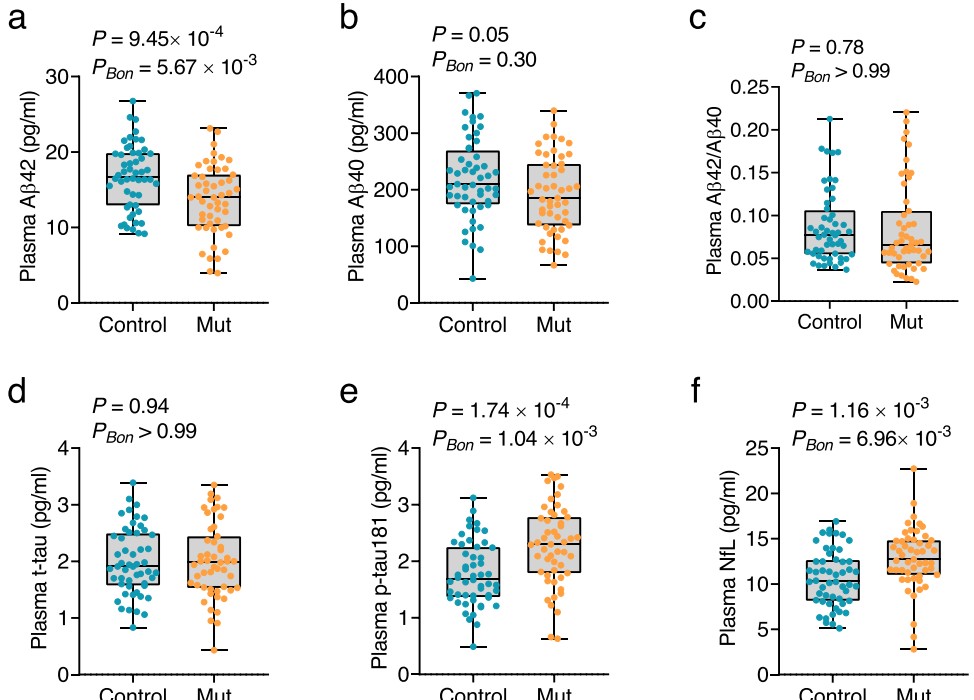

**Fig. 6 | Levels of plasma biomarkers in cohort 2.** Plasma levels of Aβ42 (**a**), Aβ40 (**b**), Aβ42/Aβ40 (**c**), t-tau (**d**), p-tau181 (**e**), and NfL (**f**) were measured in cohort 2. *P*-values and Bonferroni-corrected *P*-values ($P_{Bon}$) from two-sided Student's *t*-test comparing mutation carriers (orange dots) and normal controls (blue dots) are shown separately in each panel. Box ends represent the 25th and 75th percentiles, and the horizontal line within each box indicates the median. Whiskers extend to the upper and lower adjacent values or the most extreme points within the 1.5 × interquartile range of the 25th and 75th percentiles. *n* = 52 (control), 51 (Mut). Aβ amyloid-β, AD Alzheimer's disease, Bon Bonferroni, NfL neurofilament light chain, Mut mutation carrier, p-tau phosphorylated tau, t-tau total tau. Source data are provided as a Source Data file.

studies[47,48] have revealed that tau in CSF comprises N-terminal and mid-region species, while in blood, it predominantly exists in N-terminal forms. Post-translational modification or peripheral metabolism of plasma tau yields shorter N-terminal fragments lacking the mid-region, making them unrecognizable by classic t-tau assays[49]. This might explain the poor correlation between plasma t-tau and its CSF counterparts and the ineffective differentiating ability of the biomarker. Nonetheless, some assays targeting shorter N-terminal-bearing fragments of plasma tau showed a strong correlation with CSF t-tau and the ability to differentiate AD from controls[49]. This suggests, with appropriate antibodies, that plasma t-tau could provide added value to AD diagnosis or even prediction, which warrants confirmation in further research.

The current study had some limitations. First, although we measured Aβ42, Aβ40, and p-tau181, the main hallmarks of AD, other promising variants, such as p-tau217 and p-tau231, should also be investigated. Second, this study did not include individuals with MCI, limiting the generalizability of these results to other AD populations. Finally, validation by larger, diverse, and independent cohorts is necessary before the findings can be implemented for a broader application.

In conclusion, our study revealed that plasma levels of Aβ42, p-tau181, and NfL are reliable indicators capable of reflecting changes in CSF, and in combination, they can accurately identify preclinical AD at least 8 years prior to symptom onset in Chinese populations. Plasma levels of Aβ42, p-tau181, and NfL show promise as clinical and research screening tools, pending validation in independent and multiethnic cohorts.

## Methods
### Participants
This study complies with all relevant ethical regulations. The study protocol was approved by the Institutional Review Board of Xuanwu Hospital, Capital Medical University. Participants were not compensated and informed consent was obtained from all the participants or their legal guardians before enrollment. This study was based on a longitudinal study for prediction assessment and a cross-sectional study for validation in two independent cohorts. All the participants were Chinese. Cohort 1 consisted of participants from the China COAST[50] who were cognitively intact at baseline 8–10 years before the study (between 2012 and 2014). Blood samples were collected at baseline and follow-up; CSF samples were collected only at follow-up. Biomarkers in both blood and CSF were measured at follow-up. The diagnosis of AD was based on both the clinical and biomarker criteria. Clinical diagnosis of AD was established separately by at least two AD specialists according to the 2011 NIA-AA criteria[51]. Participants with AD must meet a biomarker criterion (CSF p-tau181/Aβ42 > 0.14) (Supplementary Fig. 2) based on our previously published data[52], which is consistent with other studies[53]. Furthermore, we used a reported CSF Aβ42 value of <500 pg/ml[54] (Supplementary Fig. 2) as another inclusion criterion of AD because low CSF Aβ42 is a critical pathological change in AD according to the ATN framework[7]. In this study, the clinical onset of the disease refers to cognitive impairment reaching the diagnostic criteria of AD. In cohort 1, participants who were cognitively intact at both baseline and follow-up were considered normal controls. Participants who were cognitively intact at baseline but developed AD at follow-up were determined to have preclinical AD at baseline. Participants who were cognitively impaired reaching the diagnostic criteria of MCI at follow-up were not included in this study.

The replication cohort (cohort 2) was recruited from the CFAN, which was previously partially published[55–57]. The participants were from families carrying mutations in known causative genes of FAD: amyloid precursor protein (*APP*), presenilin 1 (*PSEN1*), or presenilin 2 (*PSEN2*). It is believed that the age at onset tends to be accordant for a

given mutation within a family[27]; therefore, the average age at onset can be calculated within each family. EYO can be defined by subtracting the age of the individuals (young mutation carriers) in a family from the average age at onset. Additionally, EYO has been used to recruit participants with preclinical AD in our studies and others previously[56–58]. In this study, mutation carriers with an EYO of 8–10 years were included and determined to have preclinical AD. Age- and sex-matched mutation non-carriers within the families served as normal controls.

## Blood sample collection and measurement
Whole blood samples were collected into ethylenediaminetetraacetic acid tubes after a 12-h fast. Blood samples were centrifuged at 4200 g for 10 min to obtain plasma. Plasma supernatants were aliquoted and stored at −80 °C until analysis. Plasma levels of Aβ42, Aβ40, p-tau181, t-tau, and NfL were diluted fourfold and quantified using the Simoa method (Supplementary Table 3) on a fully automated HD-X Analyzer (Quanterix) according to the manufacturer's protocol. A single assay lot was used to evaluate the samples. Calibrators and quality controls were analyzed in duplicate in each run. The measurements were performed in a blinded manner.

## CSF sample collection and measurement
CSF samples were obtained through lumbar punctures and handled according to standard procedures[59]. Puncture was performed at the L3-L5 intervertebral disc spaces, and 15-milliliter CSF samples were obtained using atraumatic 20-gauge needles. Subsequently, the CSF samples were centrifuged at 2000 g for 10 min at room temperature and stored at −80∘C in polypropylene tubes. CSF levels of Aβ42, Aβ40, p-tau181, t-tau, and NfL were measured using commercially available enzyme-linked immunosorbent assay kits (Supplementary Table 4), following the manufacturer's instructions. The measurements were performed in a blinded manner.

## Statistical analysis
Data from each cohort were independently analyzed. Categorical data, such as self-reported sex and *APOE ε*4 status, were compared using the χ2 test. Numerical data, such as age, educational attainment, MMSE scores, and biomarker levels, were analyzed using Student's *t*-test or Mann-Whitney test, as appropriate. Correlation analysis was performed using a linear regression model. A binary logistic regression model was constructed to determine the predictive value of plasma biomarkers with adjustment for age, sex, educational attainment, and *APOE ε*4 status. Model accuracy was tested using ROC curve analysis. The tolerances, VIFs, eigenvalues, and condition indices were estimated to examine multicollinearity between each biomarker. The statistical significance threshold was set at $P < 0.05$, two-sided. Multiple comparisons were corrected by the Bonferroni method. The above analyses were performed using SPSS Statistics for Windows version 22 (IBM). R studio and R version 4.2.3 (package DescTools 0.99.48 and pROC 1.18.0) were used to calculate the Lin's CCC as well as AIC and to compare AUCs.

## Reporting summary
Further information on research design is available in the Nature Portfolio Reporting Summary linked to this article.

## Data availability
The de-identified raw data are available from the corresponding author Longfei Jia (longfei@mail.ccmu.edu.cn) to researchers who provide methodologically sound scientific proposals. A materials transfer and/ or data access agreement will be required for accessing shared data. Source data are provided with this paper.

## Code availability
The code used for producing the results presented in this study is available at Zenodo[60]: https://zenodo.org/record/8375702.

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

## Acknowledgements
This study was supported by the National Natural Science Foundation of China (Grant Nos. 81870825 and 82071194) to L.J., the Beijing Brain Initiative from the Beijing Municipal Science & Technology Commission (Grant No. Z201100005520016) to L.J., the Beijing Municipal Natural Science Foundation (Grant No. 7202061) to L.J., the Capital's Funds for Health Improvement and Research (Grant No.2022-2-2017) to L.J., and the STI 2030-Major Projects (Grant No. 2022ZD0211600, 2022ZD0211605) to L.J.

## Author contributions
L.J. had full access to all the data in the study and took responsibility for the integrity of the data and the accuracy of the data analysis. H.C. and L.J. designed the study. H.C., L.J., Y.P., X.F., and Z.R. contributed to the acquisition, analysis, and interpretation of data. H.C. and L.J. drafted the manuscript. H.C., L.J., Y.P., X.F., and Z.R. critically revised the manuscript for important intellectual content. Y.P., X.F., and Z.R. provided administrative, technical, and material support. All authors approved the final manuscript.

## Competing interests
The authors declare no competing interests.
