## [Peer Review File · Nature Communications]

Plasma Biomarkers Predict Alzheimer's Disease Before Clinical Onset in Chinese CohortsReviewer #1 (Remarks to the Author):

In this manuscript, Cai et al., clearly describe plasma ATN analysis of two well matched sample cohorts. They use these two cohorts to show that plasma ATN values at baseline predict cognitive decline up to 10 years later in sporadic and familial AD. The assays they use are well established, the sample cohorts are extremely well matched, reasonably well powered, and the manuscript is clear and well written. I believe this work is a solid contribution to the field, particularly as it presumably samples a quite different population (Chinese individuals recruited at Xuanwu Hospital) to most cohort studies currently published. Indeed, this aspect of study novelty should be highlighted by the Authors.

Comments:

1) The term "highly associated" and "Closely associated" are too non-specific – use the phrase significantly correlated where appropriate.

2) Race should be added to Table 1 or referred to in the Methods

3) Line 87, I would add "In cohort 2, 51 familial AD mutation carriers", so that it's clear we're not talking about ApoE

4) Was there just a single follow up for these individuals or do other datapoints for this dataset (interim cognitive testing or plasma samples) exist?

5) The lower levels of AB40 in plasma at baseline are interesting, as usually AB40 is used as a stable normalizer for peptide loss to plastics during sample collection, processing and analysing. In this sample set it is clearly not suitable for performing this role. This should be more clearly discussed in the manuscript. Is the same effect present in the CSF (AB40 lower in preAD), or is this change specific to the plasma. Why might this be? I note that the assays are different between CSF and plasma.

6) Lines 131 and 132 – Figure 4 is not in the supplement

7) Figure 4 – Looking at the axis numbering, it looks like the actual values of the ATN markers in plasma don't change much within an individual between baseline and follow-up. Could this be plotted more clearly or tested statistically (Lin's Concordance Correlation Coefficient could be of use here)? Are these values changing over 10 years, or are most of the changes happening 8-10 years before decline? If so, are these markers actually better for predicting disease onset than predicting disease course post-diagnosis? This should be discussed in the manuscript.

8) Line 152 uses the word "associated" again. This should be more specific – In a multivariate model with x,y,z as explanatory variables, AB42, pTau181 and NfL were significant contributors to the model, unlike Age, sex etc" as an example

9) Does ApoE4 significantly improve the logistic regression model at baseline? This can be tested using a likelihood ratio test.

10) Figures 5C and D are somewhat circular given the high correlation of CSF and plasma biomarkers, and the fact the CSF biomarkers were used to define the cohorts.

11) In the methods, how is disease onset defined? A drop in MMSE of X points?

12) Is there a difference in ROC performance between the different familial mutations, or do they look very similar?

13) The discussion doesn't make it clear what this study adds to the already busy field – lines 219-234 should make the novelty of this piece of work more evident

14) Lines 265 re. total tau – presumably the Innotest versus Quanterix assays use different antibodies. Mass spec is showing potentially different tau fragment species in different biofluids – this should probably also be discussed here. Is the assay used by the Framingham Heart Study the same Quanterix assay?

15) I'm not sure what lines 302-303 means, and this is important for assessing disease timecourse in cohort 2

16) Please add Quanterix assay details to eTable 1, and make it clear which are used for CSF versus plasma.

17) Has multiple test correction been performed on the p values quoted in each figure? I doubt it would change the significance, but should still be performed for each analysis.

Reviewer #2 (Remarks to the Author):

This is a very well-written and well-designed study that describes an important confirmation on how plasma biomarkers for AD-related pathologies relate to their corresponding CSF measures and how they perform diagnostically in different stages of the AD process. One criticism could be that the study lacks in novelty, but I think this is a very important confirmation in an Asian population with excellent reference tests in the form of the CSF biomarkers to compare against. All this makes the study an important and valuable contribution to the literature. In short, I have no significant criticism to any aspect of the research conducted.

Reviewer #3 (Remarks to the Author):

Cai and colleagues, studied usefulness of plasma A β 42, p-tau181 and NfL as diagnostic and prognostic biomarkers of Alzheimer's disease and their associations with the corresponding CSF biomarkers. While long follow-up time is a strength of this study, the characteristics of the included cohorts make me doubt the generalizability of the reported findings. In particular, cohort 1 appears to be highly selected with more than half of the participants who were cognitively unimpaired at baseline showing cognitive decline at follow-up. It is surprising that there were no differences in A β 42/A β 40 between the control and preAD groups. The correlations between plasma and CSF biomarkers are unexpectedly high compared to what has been reported in several previous studies that used the same plasma assays. The authors mistakenly state that "it remains unclear whether changes in peripheral blood biomarkers reflect changes in CSF levels." There is quite a number of studies demonstrating associations between CSF and plasma levels of different AD biomarkers and even some studies testing if these associations differ across different assays used for measurements of the same plasma biomarkers. Finally, the performance of the model combining baseline plasma A β 42, p-tau181 and NfL data to differentiate participants who experienced cognitive decline over time from those who did not is low to have any implications. The ROC analysis of the followup plasma data is not really meaningful considering that the AD group appears to have advanced disease (even though the exact cognitive status of participants with AD at followup is not specified).

REVIEWER COMMENTS

Reviewer #1 (Remarks to the Author):

In this manuscript, Cai et al., clearly describe plasma ATN analysis of two well matched sample cohorts. They use these two cohorts to show that plasma ATN values at baseline predict cognitive decline up to 10 years later in sporadic and familial AD. The assays they use are well established, the sample cohorts are extremely well matched, reasonably well powered, and the manuscript is clear and well written. I believe this work is a solid contribution to the field, particularly as it presumably samples a quite different population (Chinese individuals recruited at Xuanwu Hospital) to most cohort studies currently published. Indeed, this aspect of study novelty should be highlighted by the Authors.

Comments:

1) The term “highly associated” and “Closely associated” are too non-specific – use the phrase significantly correlated where appropriate.

Response: Thank you for pointing this out. We have changed the terms “highly associated” and “closely associated” to “significantly correlated.” (page 2, paragraph 1; page 6, paragraph 1; page 8, paragraph 2 and page 9, paragraph 2)

2) Race should be added to Table 1 or referred to in the Methods

Response: Thank you for pointing this out. All the participants in this study were Chinese Han. We have added this information in the ‘Methods’ section (page 14, paragraph 1)

3) Line 87, I would add “In cohort 2, 51 familial AD mutation carriers”, so that it’s clear

we're not talking about ApoE

Response: Thank you for your valuable comment. We have changed the sentence to “familial AD mutation carriers.” (page 5, paragraph 1)

4) Was there just a single follow up for these individuals or do other datapoints for this dataset (interim cognitive testing or plasma samples) exist?

Response: Thank you for your valuable comment. In this longitudinal cohort, follow-up was performed every two years, accompanied by collection of cognitive testing data and plasma samples. Using this dataset, we will soon submit a manuscript reporting the association between blood biomarkers and changes in cognition during the course of the disease.

5) The lower levels of A β 40 in plasma at baseline are interesting, as usually A β 40 is used as a stable normalizer for peptide loss to plastics during sample collection, processing and analysing. In this sample set it is clearly not suitable for performing this role. This should be more clearly discussed in the manuscript. Is the same effect present in the CSF (A β 40 lower in preAD), or is this change specific to the plasma. Why might this be? I note that the assays are different between CSF and plasma.

Response: Thank you for your valuable comment. We totally agree that A β 40 can be used as a stable normalizer during sample collection, processing, and analysis. However, in this study, the levels of A β 40 in the plasma of PreAD at baseline had a downward trend (not significant, $P > 0.05$). We added the following discussion in the revised manuscript: “A β 40 has been considered a stable normalizer during sample collection, processing, and analysis^{1,2}. However, our study observed a non-significant downward

trend in the plasma A β 40 level in participants with preclinical AD at baseline. Similar findings were previously reported in subjective cognitive decline (SCD), MCI, and A β -positive non-demented participants³⁻⁶. Studies have shown that different test methods can affect the results⁷; therefore, it is unclear whether the differences in results are due to distinct methods or even racial disparities across studies. The reason for the non-significantly lower levels of A β 40 observed in our study and others may need further investigation in international multicenter studies” (page 11, paragraph 2). For the CSF A β 40 level, unfortunately, we did not collect CSF at baseline. However, previous studies have demonstrated conflicting results regarding A β 40 levels in CSF, in which both decrease⁸ or no significant changes⁹⁻¹⁴ have been reported in MCI, SCD or A β positive non-demented participants. Concordant with the change in plasma in our study, there are also studies^{10,13,15} that found CSF A β 40 levels are non-significantly lower in MCI or A β positive non-demented participants. The discrepancies across these studies could be, again, attributed to preanalytical and analytical method variations, heterogeneity in the selected cohort, or the use of specific inclusion criteria (i.e. CSF biomarkers vs. PET).

References:

1. Hansson, O., Lehmann, S., Otto, M., Zetterberg, H. & Lewczuk, P. Advantages and disadvantages of the use of the CSF Amyloid β (A β) 42/40 ratio in the diagnosis of Alzheimer’s Disease. *Alzheimers Res. Ther.* **11**, 34 (2019).
2. Lewczuk, P. *et al.* Cerebrospinal fluid and blood biomarkers for neurodegenerative dementias: An update of the Consensus of the Task Force on Biological Markers in

- Psychiatry of the World Federation of Societies of Biological Psychiatry. *World J. Biol. Psychiatry* **19**, 244–328 (2018).
3. Stockmann, J. *et al.* Amyloid- β misfolding as a plasma biomarker indicates risk for future clinical Alzheimer's disease in individuals with subjective cognitive decline. *Alzheimers Res. Ther.* **12**, 169 (2020).
 4. Palmqvist, S. *et al.* An accurate fully automated panel of plasma biomarkers for Alzheimer's disease. *Alzheimers Dement.* **19**, 1204–1215 (2023).
 5. Xiao, Z. *et al.* Plasma biomarker profiles and the correlation with cognitive function across the clinical spectrum of Alzheimer's disease. *Alzheimers Res. Ther.* **13**, 123 (2021).
 6. Wu, X. *et al.* Development of a Plasma Biomarker Diagnostic Model Incorporating Ultrasensitive Digital Immunoassay as a Screening Strategy for Alzheimer Disease in a Chinese Population. *Clin. Chem.* **67**, 1628–1639 (2021).
 7. Janelidze, S. *et al.* Head-to-Head Comparison of 8 Plasma Amyloid- β 42/40 Assays in Alzheimer Disease. *JAMA Neurol.* **78**, 1375 (2021).
 8. Hertze, J. *et al.* Evaluation of CSF Biomarkers as Predictors of Alzheimer's Disease: A Clinical Follow-Up Study of 4.7 Years. *J. Alzheimers Dis.* **21**, 1119–1128 (2010).
 9. Eruysal, E. *et al.* Sexually Dimorphic Association of Circulating Plasminogen Activator Inhibitor-1 Levels and Body Mass Index with Cerebrospinal Fluid Biomarkers of Alzheimer's Pathology in Preclinical Alzheimer's Disease. *J. Alzheimers Dis.* **91**, 1073–1083 (2023).
 10. Hansson, O. *et al.* Prediction of Alzheimer's Disease Using the CSF A β 42/A β 40

Ratio in Patients with Mild Cognitive Impairment. *Dement. Geriatr. Cogn. Disord.* **23**, 316–320 (2007).

11. Baldeiras, I. *et al.* Addition of the A β 42/40 ratio to the cerebrospinal fluid biomarker profile increases the predictive value for underlying Alzheimer’s disease dementia in mild cognitive impairment. *Alzheimers Res. Ther.* **10**, (2018).
12. Janelidze, S. *et al.* Plasma β -amyloid in Alzheimer’s disease and vascular disease. *Sci. Rep.* **6**, 26801 (2016).
13. Keshavan, A. *et al.* Concordance of CSF measures of Alzheimer’s pathology with amyloid PET status in a preclinical cohort: A comparison of Lumipulse and established immunoassays. *Alzheimers Dement. Diagn. Assess. Amp Dis. Monit.* **12**, e12097 (2020).
14. Remnestål, J. *et al.* Association of CSF proteins with tau and amyloid β levels in asymptomatic 70-year-olds. *Alzheimers Res. Ther.* **13**, 54 (2021).
15. Gao, F. *et al.* A combination model of AD biomarkers revealed by machine learning precisely predicts Alzheimer’s dementia: China Aging and Neurodegenerative Initiative (CANDI) study. *Alzheimers Dement.* **19**, 749–760 (2023).

6) Lines 131 and 132 – Figure 4 is not in the supplement

Response: Thank you for pointing this out. We have corrected the reference “Figure 4 in the supplemental file” to “Figure 4.” (page 6, paragraph 2).

7) Figure 4 – Looking at the axis numbering, it looks like the actual values of the ATN markers in plasma don’t change much within an individual between baseline and

follow-up. Could this be plotted more clearly or tested statistically (Lin's Concordance Correlation Coefficient could be of use here)? Are these values changing over 10 years, or are most of the changes happening 8-10 years before decline? If so, are these markers actually better for predicting disease onset than predicting disease course post-diagnosis? This should be discussed in the manuscript.

Response: Thank you for your valuable comment. We have redrawn Figure 4 by adding a line with 0 intercept and 45° slope, which makes it easier to see the changes in an individual between baseline and follow-up.

We also added the following information in 'Results' section of the revised manuscript: "To test whether values of the plasma biomarkers were significantly changed within an individual between baseline and follow-up, we calculated the Lin's concordance correlation coefficient (CCC). The CCC was 0.58–0.77, and the fitted lines of the linear regression deviated from the 45° line. The means of each plasma biomarker were also significantly different between baseline and follow-up (Supplementary Table 1). These findings suggested that the biomarker levels significantly differed between baseline and follow-up" (page 6, paragraph 2 and Supplementary Table 1). We also added the following information in the 'Methods' sections: "Rstudio and R version 4.2.3 (package DescTools 0.99.48, pROC 1.18.0) were used to calculate the Lin's CCC and compare AUCs" (page 16, paragraph 2). Additionally, for the issue of whether plasma biomarkers were changing over 10 years, or most of the changes happening 8–10 years before cognitive decline, we already analyzed their changing trend in our longitudinal data (follow-up every two years) and found that biomarkers have changed consistently

over 10 years. We will explore whether the combination of plasma biomarkers can predict disease course in another article soon.

8) Line 152 uses the word “associated” again. This should be more specific – In a multivariate model with x,y,z as explanatory variables, A β 42, pTau181 and NfL were significant contributors to the model, unlike Age, sex etc” as an example

Response: Thank you for your comment. We have changed the sentence to “After adjusting for age, sex, educational attainment, and *APOE* ϵ 4 status, plasma A β 42, P-tau181, and NfL were significant contributors to the occurrence of AD. Age, sex, and educational attainment were not statistically significant ($P > 0.05$) and were excluded from the final model” (page 7, paragraph 1).

9) Does ApoE4 significantly improve the logistic regression model at baseline? This can be tested using a likelihood ratio test.

Response: Thank you for your comment. We have added the results as follows: “The likelihood ratio test showed that adding *APOE* ϵ 4 status to the models significantly improved the disease prediction accuracy at baseline ($P < 0.05$) but not at follow-up ($P > 0.05$). The DeLong test further confirmed that adding *APOE* ϵ 4 status significantly increased the AUC of the ROC analysis at baseline (AUC = 0.81, 95% CI: 0.75-0.86, $P < 0.05$; Figure 5a) but not at follow-up (AUC = 0.99, 95% CI: 0.98-0.99, $P > 0.05$; Figure 5c).” (page 7, paragraph 1).

10) Figures 5C and D are somewhat circular given the high correlation of CSF and plasma biomarkers, and the fact the CSF biomarkers were used to define the cohorts.

Response: We totally agree with you that circular proof is invalid in biomarker research.

For example, we cannot study the diagnostic capacity of CSF A β 42 in participants recruited by CSF A β 42 levels. In our study, we aimed to study plasma biomarkers for a more accurate diagnosis of AD with strict inclusion criteria; we used CSF biomarkers (A β <500 pg/mL and P-tau/A β 42>0.14) as inclusion criteria of AD. Other studies in this field have also used A β -PET or CSF A β as inclusion criteria to study the diagnostic value of plasma A β 42. In the current study, the values of R² in the correlation analyses of A β 42, P-tau181, and NfL were 0.53–0.66, indicating that plasma biomarkers can only reflect 53%-66% of their CSF counterparts, making the models in Figures 5C and D not circular proof. For example, in dataset 1 at follow-up, when using CSF A β 42=500 pg/mL as the cutoff value, we can completely distinguish AD from controls (AUC=1). However, when using plasma A β 42 or plasma P-tau181 as a single biomarker to identify AD from controls, their performance is suboptimal(AUC: A β 42=0.9, P-tau181=0.9), while the combination of plasma A β 42, P-tau181 and NfL achieved an AUC of 0.99 (shown in Figures 5c and d). These data suggest that the diagnostic models may not be circular.

11) In the methods, how is disease onset defined? A drop in MMSE of X points?

Response: Thank you for your valuable comment. We apologize for this confusion. Onset refers to cognitive impairment that meets the diagnostic criteria for AD (NIA-AA criteria). We have added the following information in the revised manuscript: “The onset of the disease refers to cognitive impairment reaching the diagnostic criteria of AD, while cognitive impairment reaching the diagnostic criteria of MCI was not counted as onset until cognitive decline meets the diagnostic criteria of AD” (page 14,

paragraph 2).

12) Is there a difference in ROC performance between the different familial mutations, or do they look very similar?

Response: Thank you for your valuable comment. We separately performed ROC analyses in *APP*, *PSEN1*, or *PSEN2* mutation carriers and controls. The AUCs of each group were very similar, ranging between 0.78–0.80. We have added these data to the Supplementary Materials (Supplementary Fig. 1).

13) The discussion doesn't make it clear what this study adds to the already busy field – lines 219-234 should make the novelty of this piece of work more evident

Response: Thank you for your valuable comment. We have added the following discussion in the revised manuscript: “Although many studies have investigated the relationship between plasma biomarkers and AD, conducting research across ethnic and racial diversities is crucial, as these factors may influence the assessment of biological risks associated with AD¹⁻⁴. This study was conducted among the Asian race, to be specific, a Chinese Han population, which reinforces previous findings from European and American populations⁵⁻⁷. Our findings further add to the evidence of plasma biomarkers as reliable prognostic candidates for AD and highlight their promising roles in future clinical practice and research.” (page 10, paragraph 1).

References:

1. Schindler, S. E. *et al.* Effect of Race on Prediction of Brain Amyloidosis by Plasma A β 42/A β 40, Phosphorylated Tau, and Neurofilament Light. *Neurology* **99**, e245–e257 (2022).

2. Grewal, R. *et al.* Identifying biomarkers of dementia prevalent among amnesic mild cognitively impaired ethnic female patients. *Alzheimers Res. Ther.* **8**, 43 (2016).
3. Babulal, G. M. *et al.* Perspectives on ethnic and racial disparities in Alzheimer's disease and related dementias: Update and areas of immediate need. *Alzheimers Dement.* **15**, 292–312 (2019).
4. Howell, J. C. *et al.* Race modifies the relationship between cognition and Alzheimer's disease cerebrospinal fluid biomarkers. *Alzheimers Res. Ther.* **9**, (2017).
5. Palmqvist, S. *et al.* Prediction of future Alzheimer's disease dementia using plasma phospho-tau combined with other accessible measures. *Nat. Med.* **27**, 1034–1042 (2021).
6. Palmqvist, S. *et al.* An accurate fully automated panel of plasma biomarkers for Alzheimer's disease. *Alzheimers Dement.* **19**, 1204–1215 (2023).
7. Cullen, N. C. *et al.* Plasma biomarkers of Alzheimer's disease improve prediction of cognitive decline in cognitively unimpaired elderly populations. *Nat. Commun.* **12**, 3555 (2021).

14) Lines 265 re. total tau – presumably the Innotest versus Quanterix assays use different antibodies. Mass spec is showing potentially different tau fragment species in different biofluids – this should probably also be discussed here. Is the assay used by the Framingham Heart Study the same Quanterix assay?

Response: Thank you for your valuable comment. We totally agree that the T-tau assays

we used in the study targeted different epitopes of tau. We have added the following discussion in the revised manuscript: “Furthermore, we used Innostest method for measurements of CSF T-tau, and Quanterix assays for plasma T-tau. The Innostest test for CSF T-tau is directed at mid-region epitopes, while the Quanterix Neurology 3-plex kit for plasma T-tau targets N-terminal-to-mid-region epitopes, which targets epitopes identical to the Quanterix Tau 2.0 kit used in the Framingham Heart Study¹. Mass spectrometry studies^{2,3} have revealed that tau in CSF comprises N-terminal and mid-region species, while in blood, it predominantly exists in N-terminal forms. Post-translational modification or peripheral metabolism of plasma tau yields shorter N-terminal fragments lacking the mid-region, making them unrecognizable by classic T-tau assays⁴. This might explain the poor correlation between plasma T-tau and its CSF counterparts and the ineffective differentiating ability of the biomarker. Nonetheless, some novel assays targeting shorter N-terminal-bearing fragments of plasma tau showed a strong correlation with CSF T-tau and the ability to differentiate AD from controls⁴. This suggests, with appropriate antibodies, that plasma T-tau could provide added value to AD diagnosis or even prediction, which warrants confirmation in further research.” (page 12, paragraph 2)

References:

1. Pase, M. P. *et al.* Assessment of Plasma Total Tau Level as a Predictive Biomarker for Dementia and Related Endophenotypes. *JAMA Neurol.* **76**, 598–606 (2019).
2. Barthélemy, N. R., Horie, K., Sato, C. & Bateman, R. J. Blood plasma phosphorylated-tau isoforms track CNS change in Alzheimer’s disease. *J. Exp. Med.*

217, e20200861 (2020).

3. Cicognola, C. *et al.* Novel tau fragments in cerebrospinal fluid: relation to tangle pathology and cognitive decline in Alzheimer's disease. *Acta Neuropathol. (Berl.)* **137**, 279–296 (2019).
4. Snellman, A. *et al.* N-terminal and mid-region tau fragments as fluid biomarkers in neurological diseases. *Brain* **145**, 2834–2848 (2022).

15) I'm not sure what lines 302-303 means, and this is important for assessing disease timecourse in cohort 2

Response: Thank you for your valuable comment. We apologize for this confusion. We have added the following information in the revised manuscript: “It is believed that the age at onset tends to be accordant for a given mutation within a family¹; therefore, the average age at onset can be calculated within each family. EYO can be defined by subtracting the age of the individuals (young mutation carriers) in a family from the average age at onset. Additionally, EYO has been used to recruit participants with preclinical AD in our studies and others previously²⁻⁴.”(page 15, paragraph 1)

References:

1. Moulder, K. L. *et al.* Dominantly Inherited Alzheimer Network: facilitating research and clinical trials. *Alzheimers Res. Ther.* **5**, 48 (2013).
2. Preische, O. *et al.* Serum neurofilament dynamics predicts neurodegeneration and clinical progression in presymptomatic Alzheimer's disease. *Nat. Med.* **25**, 277–283 (2019).

3. Jia, L. *et al.* Blood neuro-exosomal synaptic proteins predict Alzheimer's disease at the asymptomatic stage. *Alzheimers Dement.* **17**, 49–60 (2021).
4. Jia, L. *et al.* Exosomal MicroRNA-Based Predictive Model for Preclinical Alzheimer's Disease: A Multicenter Study. *Biol. Psychiatry* **92**, 44–53 (2022).

16) Please add Quanterix assay details to eTable 1, and make it clear which are used for CSF versus plasma.

Response: Thank you for your comment. Details of the Quanterix assay have been added to Supplementary Table 2. Quanterix assays were used to measure plasma biomarkers, while INNOTEST ELISA kits were used to measure CSF biomarkers. We have clarified these in the tables (Supplementary Tables 2 and 3).

17) Has multiple test correction been performed on the p values quoted in each figure? I doubt it would change the significance, but should still be performed for each analysis.

Response: Thank you for your valuable comment. We used the Bonferroni method for multiple test corrections. Because six hypothesis tests were conducted, each *P* value resulting from the t-test was multiplied by six, and the results were then compared with 0.05. We have added this information to the revised manuscript, revised figures and accompanying legends (page 16, paragraph 1; Figures 1, 3 and 6 and accompanying legends).

Reviewer #2 (Remarks to the Author):

This is a very well-written and well-designed study that describes an important confirmation on how plasma biomarkers for AD-related pathologies relate to their

corresponding CSF measures and how they perform diagnostically in different stages of the AD process. One criticism could be that the study lacks in novelty, but I think this is a very important confirmation in an Asian population with excellent reference tests in the form of the CSF biomarkers to compare against. All this makes the study an important and valuable contribution to the literature. In short, I have no significant criticism to any aspect of the research conducted.

Response: Thank you for your valuable comment. To further improve our manuscript, we have added the following discussion in the revised manuscript: “Although many studies have investigated the relationship between plasma biomarkers and AD, conducting research across ethnic and racial diversities is crucial, as these factors may influence the assessment of biological risks associated with AD¹⁻⁴. This study was conducted among the Asian race, to be specific, a Chinese Han population, which reinforces previous findings from European and American populations⁵⁻⁷. Our findings further add to the evidence of plasma biomarkers as reliable prognostic candidates for AD and highlight their promising roles in future clinical practice and research.” (page 10, paragraph 1).

References:

1. Schindler, S. E. *et al.* Effect of Race on Prediction of Brain Amyloidosis by Plasma A β 42/A β 40, Phosphorylated Tau, and Neurofilament Light. *Neurology* **99**, e245–e257 (2022).
2. Grewal, R. *et al.* Identifying biomarkers of dementia prevalent among amnesic mild cognitively impaired ethnic female patients. *Alzheimers Res. Ther.* **8**, 43

- (2016).
3. Babulal, G. M. *et al.* Perspectives on ethnic and racial disparities in Alzheimer's disease and related dementias: Update and areas of immediate need. *Alzheimers Dement.* **15**, 292–312 (2019).
 4. Howell, J. C. *et al.* Race modifies the relationship between cognition and Alzheimer's disease cerebrospinal fluid biomarkers. *Alzheimers Res. Ther.* **9**, (2017).
 5. Palmqvist, S. *et al.* Prediction of future Alzheimer's disease dementia using plasma phospho-tau combined with other accessible measures. *Nat. Med.* **27**, 1034–1042 (2021).
 6. Palmqvist, S. *et al.* An accurate fully automated panel of plasma biomarkers for Alzheimer's disease. *Alzheimers Dement.* **19**, 1204–1215 (2023).
 7. Cullen, N. C. *et al.* Plasma biomarkers of Alzheimer's disease improve prediction of cognitive decline in cognitively unimpaired elderly populations. *Nat. Commun.* **12**, 3555 (2021).

Reviewer #3 (Remarks to the Author):

Cai and colleagues, studied usefulness of plasma A β 42, p-tau181 and NfL as diagnostic and prognostic biomarkers of Alzheimer's disease and their associations with the corresponding CSF biomarkers. While long follow-up time is a strength of this study, the characteristics of the included cohorts make me doubt the generalizability of the reported findings. In particular, cohort 1 appears to be highly selected with more than

half of the participants who were cognitively unimpaired at baseline showing cognitive decline at follow-up. It is surprising that there were no differences in A β 42/A β 40 between the control and preAD groups. The correlations between plasma and CSF biomarkers are unexpectedly high compared to what has been reported in several previous studies that used the same plasma assays. The authors mistakenly state that "it remains unclear whether changes in peripheral blood biomarkers reflect changes in CSF levels." There is quite a number of studies demonstrating associations between CSF and plasma levels of different AD biomarkers and even some studies testing if these associations differ across different assays used for measurements of the same plasma biomarkers. Finally, the performance of the model combining baseline plasma A β 42, p-tau181 and NfL data to differentiate participants who experienced cognitive decline over time from those who did not is low to have any implications. The ROC analysis of the followup plasma data is not really meaningful considering that the AD group appears to have advanced disease (even though the exact cognitive status of participants with AD at followup is not specified).

Response:

Thank you for your valuable comments. We have carefully made point by point response to your comments as follows:

1. In particular, cohort 1 appears to be highly selected with more than half of the participants who were cognitively unimpaired at baseline showing cognitive decline at follow-up.

Response: We apologize for this confusion. The participants with AD and controls in

cohort 1 were recruited from a much larger longitudinal cohort, with up to 10 years of follow-up. All the participants in the longitudinal cohort were cognitively intact at baseline. We included participants who met the AD diagnosis at follow-up from this cohort in the current study as the patient group, and the participants who remained cognitively intact as the control group. Participants who were cognitively intact at baseline, but later developed AD at present were assessed to have preclinical AD at baseline. After age- and sex-matching, 126 patients with AD and 123 controls were selected. We used SPSS for random selection under age- and sex-matching. This case-control study design aimed to compare biomarkers between participants who are always cognitively normal during follow-up and those who are diagnosed with AD at present after follow-up. Therefore, this matching process produced nearly half of the cognitive decline in cohort 1 at follow-up.

2. It is surprising that there were no differences in A β 42/A β 40 between the control and preAD groups.

Response: Thank you for your valuable comment. We totally agree that A β 42/A β 40 may expectedly change in the preAD group. To address this issue, we double-checked the data and found no errors. After a systematic literature review, we noted that studies on plasma A β 42/A β 40 in PreAD or MCI are not consistent. We added the following discussion in the revised manuscript: “As a hallmark of AD, plasma A β 42/A β 40 may be altered in the preclinical stage of AD. However, no significant difference was detected in the plasma levels of A β 42/A β 40 between participants with preclinical AD and controls in this study. Previous studies examining plasma A β 42/A β 40 in

participants with preclinical AD or MCI have reported inconsistent findings. For example, some studies showed no significant change in plasma A β 42/A β 40 levels in participants with preclinical AD or MCI¹⁻⁶, which is similar to our study, while others showed that plasma A β 42/A β 40 levels were reduced in these individuals compared with controls⁷⁻⁸. This is an intriguing issue to be further explored in international multicenter studies.” (page 11, paragraph 1)

References:

1. Pan, F.-F. *et al.* Non-linear Character of Plasma Amyloid Beta Over the Course of Cognitive Decline in Alzheimer’s Continuum. *Front. Aging Neurosci.* **14**, 832700 (2022).
2. Wu, X. *et al.* Development of a Plasma Biomarker Diagnostic Model Incorporating Ultrasensitive Digital Immunoassay as a Screening Strategy for Alzheimer Disease in a Chinese Population. *Clin. Chem.* **67**, 1628–1639 (2021).
3. Silva-Spínola, A. *et al.* Blood biomarkers in mild cognitive impairment patients: Relationship between analytes and progression to Alzheimer disease dementia. *Eur. J. Neurol.* (2023) doi:10.1111/ene.15762.
4. Gao, F. *et al.* A combination model of AD biomarkers revealed by machine learning precisely predicts Alzheimer’s dementia: China Aging and Neurodegenerative Initiative (CANDI) study. *Alzheimers Dement.* **19**, 749–760 (2023).
5. Simrén, J. *et al.* The diagnostic and prognostic capabilities of plasma biomarkers in Alzheimer’s disease. *Alzheimers Dement. J. Alzheimers Assoc.* **17**, 1145–1156 (2021).

6. Palmqvist, S. *et al.* An accurate fully automated panel of plasma biomarkers for Alzheimer's disease. *Alzheimers Dement.* **19**, 1204–1215 (2023).
7. Doecke, J. D. *et al.* Total A β_{42} /A β_{40} ratio in plasma predicts amyloid-PET status, independent of clinical AD diagnosis. *Neurology* **94**, e1580–e1591 (2020).
8. Stockmann, J. *et al.* Amyloid- β misfolding as a plasma biomarker indicates risk for future clinical Alzheimer's disease in individuals with subjective cognitive decline. *Alzheimers Res. Ther.* **12**, 169 (2020).

3. The correlations between plasma and CSF biomarkers are unexpectedly high compared to what has been reported in several previous studies that used the same plasma assays. The authors mistakenly state that "it remains unclear whether changes in peripheral blood biomarkers reflect changes in CSF levels." There is quite a number of studies demonstrating associations between CSF and plasma levels of different AD biomarkers and even some studies testing if these associations differ across different assays used for measurements of the same plasma biomarkers.

Response: Thank you for your valuable comment. We fully agree that there are several studies demonstrating associations between CSF and plasma levels. We have removed the sentence "It remains unclear whether changes in peripheral blood biomarkers reflect changes in CSF levels" and other similar expressions from our revised manuscript. To determine the reason why the correlations between plasma and CSF biomarkers in our study were higher compared to the reported data of previous studies, we rechecked the data and reperformed the calculations and found no errors. After a systematic literature

review, we noted that the correlations between plasma and CSF biomarkers are not consistent in this field. For comparison with other studies, we calculated the values of the r and β coefficients. We have added the following discussion to the supplementary materials: “According to the literature in this field, although it has been shown that about half of the CSF proteins are moderately related to their plasma counterparts¹, the correlations between plasma and CSF biomarkers are not exactly the same in previous studies. For instance, in one study² on P-tau181, the β coefficient was 0.73 in the correlation analyses between plasma and CSF P-tau181, which is completely consistent with our data ($\beta=0.73$, correlation between plasma and CSF P-tau181), while the values of r in another study³ were 0.7055 (plasma vs. CSF) and 0.7937 (serum vs. CSF), which were higher or lower than ours (our $r=0.73$). In other studies⁴⁻⁸, r values ranged between 0.30–0.65, indicating that the degree of correlation between plasma and CSF P-tau181 was not consistent across studies. A similar situation exists in the correlation between plasma and CSF NfL; for instance, some studies^{9,10} ($r=0.72$ or 0.78) showed r values similar to ours (our $r=0.77$), some ($r =0.86$)⁷ higher than ours, and some ($r =0.64$ or 0.59)^{11,12} lower than ours. For A β 42, our results were higher than those of previous studies (our $r=0.81$ vs. others $r=0.35-0.41$)^{13,14}. We speculate that this may be partly due to our strict inclusion criteria of AD involving CSF biomarkers (P-tau/A β 42>0.14 and A β <500 pg/mL). Additionally, results of studies on the correlation between plasma markers and CSF were quite different, even in the same article. For example, one study² showed that a correlation analysis of P-tau181 was $\beta=0.73$ in dataset one, and $\beta=0.52$ in dataset two, while another study¹⁵ showed the results for NfL, $R^2=0.27$ in

BioFINDER-1 vs. $R^2=0.49$ in BioFINDER-2, for GFAP, $R^2=0.13$ in BioFINDER-1 vs. $R^2=0.43$ in BioFINDER-2. The above data indicate that in the correlation analysis between plasma and CSF, the correlation strength of biomarkers varies among studies or even in different datasets of the same study. The underlying reasons for the inconsistency remain unclear and could be attributed to methodological variations, heterogeneity in the selected cohort, or the use of specific inclusion criteria (i.e. CSF biomarkers vs. PET). However, it also suggests that these plasma biomarkers can partially reflect their counterpart levels in CSF, which to some extent enhances confidence in using plasma biomarkers to aid the diagnosis of AD.” (Supplementary discussion)

References:

1. Whelan, C. D. *et al.* Multiplex proteomics identifies novel CSF and plasma biomarkers of early Alzheimer’s disease. *Acta Neuropathol. Commun.* **7**, 169 (2019).
2. Janelidze, S. *et al.* Plasma P-tau181 in Alzheimer’s disease: relationship to other biomarkers, differential diagnosis, neuropathology and longitudinal progression to Alzheimer’s dementia. *Nat. Med.* **26**, 379–386 (2020).
3. Karikari, T. K. *et al.* Blood phosphorylated tau 181 as a biomarker for Alzheimer’s disease: a diagnostic performance and prediction modelling study using data from four prospective cohorts. *Lancet Neurol.* **19**, 422–433 (2020).
4. Altomare, D. *et al.* Plasma biomarkers for Alzheimer’s disease: a field-test in a memory clinic. *J. Neurol. Neurosurg. Psychiatry* jnnp-2022-330619 (2023)

doi:10.1136/jnnp-2022-330619.

5. Tropea, T. F. *et al.* Plasma phosphorylated tau181 predicts cognitive and functional decline. *Ann. Clin. Transl. Neurol.* **10**, 18–31 (2023).
6. Karikari, T. K. *et al.* Diagnostic performance and prediction of clinical progression of plasma phospho-tau181 in the Alzheimer’s Disease Neuroimaging Initiative. *Mol. Psychiatry* **26**, 429–442 (2021).
7. Álvarez-Sánchez, L. *et al.* Assessment of Plasma and Cerebrospinal Fluid Biomarkers in Different Stages of Alzheimer’s Disease and Frontotemporal Dementia. *Int. J. Mol. Sci.* **24**, 1226 (2023).
8. Palmqvist, S. *et al.* An accurate fully automated panel of plasma biomarkers for Alzheimer’s disease. *Alzheimers Dement.* **19**, 1204–1215 (2023).
9. Alagaratnam, J. *et al.* Correlation between CSF and blood neurofilament light chain protein: a systematic review and meta-analysis. *BMJ Neurol. Open* **3**, e000143 (2021).
10. Alcolea, D. *et al.* Use of plasma biomarkers for AT(N) classification of neurodegenerative dementias. *J. Neurol. Neurosurg. Psychiatry* **92**, 1206–1214 (2021).
11. Parvizi, T. *et al.* Real-world applicability of glial fibrillary acidic protein and neurofilament light chain in Alzheimer’s disease. *Front. Aging Neurosci.* **14**, 887498 (2022).
12. Mattsson, N., Andreasson, U., Zetterberg, H., Blennow, K., & Alzheimer’s Disease Neuroimaging Initiative. Association of Plasma Neurofilament Light With

Neurodegeneration in Patients With Alzheimer Disease. *JAMA Neurol.* **74**, 557 (2017).

13. Nakamura, A. *et al.* High performance plasma amyloid- β biomarkers for Alzheimer's disease. *Nature* **554**, 249–254 (2018).

14. Klafki, H.-W. *et al.* Diagnostic performance of automated plasma amyloid- β assays combined with pre-analytical immunoprecipitation. *Alzheimers Res. Ther.* **14**, 127 (2022).

15. Pichet Binette, A. *et al.* Confounding factors of Alzheimer's disease plasma biomarkers and their impact on clinical performance. *Alzheimers Dement.* **19**, 1403–1414 (2023).

4. Finally, the performance of the model combining baseline plasma A β 42, p-tau181 and NfL data to differentiate participants who experienced cognitive decline over time from those who did not is low to have any implications. The ROC analysis of the followup plasma data is not really meaningful considering that the AD group appears to have advanced disease (even though the exact cognitive status of participants with AD at followup is not specified).

Response: Thank you for your valuable comment. The AUC of our model combining baseline plasma A β 42, p-tau181, and NfL to identify the PreAD 8–10 years before the onset of disease was 0.79. We totally agree that this value was relatively low, although other researchers believe that 0.7–0.8 is considered as “acceptable” and 0.8–0.9 is considered as “excellent”^{1,2}. Considering that it is truly difficult to distinguish AD early

at the asymptomatic stage (8–10 years before onset), our AUC of approximately 0.8 may be acceptable. In our previous studies, when combining plasma biomarkers to identify PreAD 5–7 years before the onset of disease, the AUC was 0.87–0.89³ or 0.85–0.88⁴. However, in the current study, when the timepoint was advanced to 8–10 years before onset, the AUC of our model decreased to 0.79. In another study, when the timepoint was advanced to 0–9 years before onset, the combined biomarkers showed an AUC of 0.791⁵, which is very similar to our results. Furthermore, when the timepoint was advanced to 9–17 years, the AUC decreased to 0.71⁵. Our work and others showed that it is hard to predict AD about 10 years or more before the onset of the disease, and in this case, an AUC close to 0.8 seems reasonable.

1. Mandrekar, J. N. Receiver Operating Characteristic Curve in Diagnostic Test Assessment. *J. Thorac. Oncol.* **5**, 1315–1316 (2010).
2. Agarwal, G. *et al.* Choosing the most appropriate existing type 2 diabetes risk assessment tool for use in the Philippines: a case-control study with an urban Filipino population. *BMC Public Health* **19**, 1169 (2019).
3. Jia, L. *et al.* Blood neuro-exosomal synaptic proteins predict Alzheimer’s disease at the asymptomatic stage. *Alzheimers Dement.* **17**, 49–60 (2021).
4. Jia, L. *et al.* Exosomal MicroRNA-Based Predictive Model for Preclinical Alzheimer’s Disease: A Multicenter Study. *Biol. Psychiatry* **92**, 44–53 (2022).
5. Stocker, H. *et al.* Association of plasma biomarkers, p-tau181, glial fibrillary acidic protein, and neurofilament light, with intermediate and long-term clinical Alzheimer’s disease risk: Results from a prospective cohort followed over 17 years.

Alzheimers Dement. **19**, 25–35 (2023).

Reviewer #1 (Remarks to the Author):

Cai et al have provided a comprehensive and detailed rebuttal, and have addressed all of my concerns. I am happy for this to go forwards for publication.

Reviewer #3 (Remarks to the Author):

I still have significant concerns about the generalizability of the findings and their implications for implementation of plasma biomarkers in clinical practice.

1. Recent evidence indicate substantial variability in the performance of different plasma A β assays as well as different p-tau variants and assays. The question that needs to be addressed now is whether there is an added value of combining the best performing plasma biomarkers such as for example IPMS-based A β 42/40 and p-tau217. Several reports have suggested inferior accuracies of plasma A β and p-tau181 assays used in the current study compared with IPMS-based methods and even compared with best performing immunoassays. Consequently, the usefulness of the findings of the present study is questionable considering that it is highly unlikely that these suboptimal assays would be implemented in clinical practice in the future.

2. The authors have not sufficiently addressed my comments regarding no difference in A β 42/A β 40 between the control and preAD groups and unexpectedly high correlations between plasma and CSF biomarkers in their study.

Previous reports using large cohorts have demonstrated reduced levels of A β 42/40 in CU amyloid positive (A+) individuals compared to CU A-. Some examples are Palmqvist et al. (JAMA Neuro, 2019), Schindler et al. (Neurology, 2019) and, in fact, Palmqvist et al. (Alzheimers Dement 2023, supplementary table 6) which the author claim showed the opposite. Furthermore, the authors cite other papers to support their statement that "some studies showed no significant change in plasma A β 42/A β 40 levels in participants with preclinical AD or MCI". However, studies by Simrén et al., Silva-Spínola et al., Wu et al. and Gao et al. did not include comparisons of plasma A β 42/40 between CU participants without AD and those with preclinical AD. It is not clear to me why the authors discuss studies including patients with clinical diagnosis of MCI who likely have cognitive impairment due to different etiologies. The results are inconsistent in this patient population because plasma A β 42/40 is expected to change in response to abnormal accumulation of A β in the brain only in patients with MCI due to AD and not in MCI with non-AD etiology. Lower plasma A β 42/40 was seen in MCI A+ than in MCI in several papers (e.g., Palmqvist et al., JAMA Neuro 2019; Palmqvist et al., Alzheimers Dement 2023).

The reported correlations between plasma and CSF A β 42 measured with the same Simoa assay and with more accurate MS-based or fully automated methods are much lower compared with the correlations if this study. The authors should clarify in more detail how they think this discrepancy in results could be explained by "strict inclusion criteria of AD".

The associations between CSF and plasma p-tau differ across p-tau variants and assays (Janelidze et al. Brain 2022; Ashton et al., Alzheimers Dement 2022). Generally, p-tau217 shows higher correlations between CSF and plasma levels than p-tau181. At the same time, there is variability in the correlations between CSF and plasma p-tau181 measured using different assays. Therefore, it is important to compare the p-tau181 results from the present study to previous work where p-tau181 was quantified using the same assay while comparisons to other p-tau forms or other p-tau181 assays are irrelevant. For example, a recent report (Ashton et al., Alzheimers Dement 2022) have demonstrated clearly lower correlations between CSF and plasma p-tau181 (R=0.52) measured using the same Quanterix Simoa assay.

3. The conclusion in the last sentence of the abstract is not supported by the results. Since CSF data were not available at baseline and the accuracy of CSF biomarkers to predict AD dementia was not assessed, it is impossible to determine whether plasma biomarkers could substitute CSF biomarkers.

4. How often did study participant undergo clinical assessments? Could the authors rule out that individuals in the preAD groups progressed to AD dementia prior to 8-10y followup?

5. Presentation of statistical results could be further improved.

(a) Please reports odds ratio, 95% CI, AIC values and p values for models including individual biomarkers or their combination as well as AUC (95% CI) of individual biomarkers

(b) Were AUCs of individual biomarker significantly different AUC of the combined model

(c) The authors should consider including all ROC curves for the same outcome in one plot

(d) In the scatterplots, please highlight in different color the control and preAD groups

REVIEWER COMMENTS

Reviewer #1 (Remarks to the Author):

Cai et al have provided a comprehensive and detailed rebuttal, and have addressed all of my concerns. I am happy for this to go forwards for publication.

Response: Thank you very much for your constructive comments concerning our manuscript. We appreciate the careful reading of our manuscript and valuable suggestions to revise and improve our paper.

Reviewer #3 (Remarks to the Author):

I still have significant concerns about the generalizability of the findings and their implications for implementation of plasma biomarkers in clinical practice.

1. Recent evidence indicate substantial variability in the performance of different plasma A β assays as well as different p-tau variants and assays. The question that needs to be addressed now is whether there is an added value of combining the best performing plasma biomarkers, such as for example IPMS-based A β 42/40 and p-tau217. Several reports have suggested inferior accuracies of plasma A β and p-tau181 assays used in the current study compared with IPMS-based methods and even compared with best performing immunoassays. Consequently, the usefulness of the findings of the present study is questionable considering that it is highly unlikely that these suboptimal assays would be implemented in clinical practice in the future.

Response: Thank you for your comment. The generalizability of our findings and their application in clinical practice are as follows:

1) In this study, we used Simoa method to detect plasma A β 42, P-tau181, and NfL, and confirmed their diagnostic capacity for AD. Simoa has emerged as an improved immunoassay technique that

is capable of detecting proteins at the femtomolar level¹. Numerous studies have indicated that the Simoa method performs accurately in AD²⁻⁶. This preference is further supported by the recent FDA Breakthrough Device designation of Quanterix Simoa P-tau181⁷. Moreover, Simoa P-tau181 has been implemented in many ongoing clinical trials as an aid for participant inclusion or as a monitoring marker for therapeutic effects, such as in trials of lecanemab⁸, aducanumab⁹, and simufilam¹⁰. However, more evidence is required before the aforementioned method can be extended to clinical applications and incorporated into guidelines. For example, current large-sample longitudinal data are mostly from European and American countries, lacking sufficient evidence concerning Asian race. However, our study provides data from independent Asian populations, contributing additional evidence to support the universal application of this particular approach.

2) We agree that immunoprecipitation-mass spectrometry (IP-MS)-based method demonstrates superior accuracy compared with the immunoassays used in this study. Furthermore, IP-MS is a powerful research tool that can identify and characterize protein-protein interactions by isolating proteins using antibodies and analyzing them by mass spectrometry¹¹. Additionally, the currently available IP-MS methods are relatively time-consuming and complex, requiring access to sophisticated mass spectrometry instruments and specialized expertise, which limits their use in routine clinical care and large-scale studies. Although IP-MS has been reported to exhibit high accuracy^{12,13}, the relative simplicity, cost-effectiveness, and sufficient diagnostic or predictive capabilities of the Simoa method make it a preferable option in clinical settings. The availability of cost-effective and sufficiently accurate diagnostic methods continues to hold significance in the fields of disease diagnosis and prognosis. In addition, the requirements for the diagnostic performance of blood-based biomarkers should be set according to the intended purpose of testing.

For instance, a test used in a clinical trial to screen participants for amyloid positivity before amyloid positron emission tomography (PET) does not need to be as accurate as a test intended to replace amyloid PET. A relatively crude marker that could be optimized to minimize false negatives would significantly reduce the number of individuals that might require cerebrospinal fluid (CSF) analysis or PET imaging, thus streamlining participant selection for clinical trials¹⁴. Careful consideration of the risks, costs, and benefits that are associated with incremental or marginal improvements in diagnostic performance is crucial^{14,15}. Moreover, if a combination of biomarkers is necessary, using a multiple-marker panel with a single assay seems more feasible in clinical practice than using IP-MS method which may need several experiments to collect each biomarker by immunoprecipitation. In this context, the findings from our study are valuable as they further support the prognostic performance of plasma A β 42, P-tau 181, and NfL measured using a relatively simple immunoassay method by providing data from an independent Asian population with follow-up.

3) To assess whether plasma biomarkers in the preclinical phase can predict the development of AD, the study recruited participants from a longitudinal cohort who were cognitively intact 8-10 years before this study and retrospectively determined their baseline diagnosis using the information at follow-up. The follow-up period is relatively long and could provide an opportunity to predict AD.

2. The authors have not sufficiently addressed my comments regarding no difference in A β 42/A β 40 between the control and preAD groups and unexpectedly high correlations between plasma and CSF biomarkers in their study.

Previous reports using large cohorts have demonstrated reduced levels of A β 42/40 in CU amyloid-positive (A+) individuals compared to CU A-. Some examples are Palmqvist et al. (JAMA Neuro, 2019), Schindler et al. (Neurology, 2019) and, in fact, Palmqvist et al. (Alzheimers Dement 2023, supplementary table 6) which the author claim showed the opposite. Furthermore, the authors cite other papers to support their statement that "some studies showed no significant change in plasma A β 42/A β 40 levels in participants with preclinical AD or MCI". However, studies by Simrén et al., Silva-Spínola et al., Wu et al., and Gao et al. did not include comparisons of plasma A β 42/40 between CU participants without AD and those with preclinical AD. It is not clear to me why the authors discuss studies including patients with clinical diagnosis of MCI who likely have cognitive impairment due to different etiologies. The results are inconsistent in this patient population because plasma A β 42/40 is expected to change in response to abnormal accumulation of A β in the brain only in patients with MCI due to AD and not in MCI with non-AD etiology. Lower plasma A β 42/40 was seen in MCI A+ than in MCI in several papers (e.g., Palmqvist et al., JAMA Neuro 2019; Palmqvist et al., Alzheimers Dement 2023).

The reported correlations between plasma and CSF A β 42 measured with the same Simoa assay and with more accurate MS-based or fully automated methods are much lower compared with the correlations in this study. The authors should clarify in more detail how they think this discrepancy in results could be explained by "strict inclusion criteria of AD".

The associations between CSF and plasma p-tau differ across p-tau variants and assays (Janelidze et al. Brain 2022; Ashton et al., Alzheimers Dement 2022). Generally, p-tau217 shows higher correlations between CSF and plasma levels than p-tau181. At the same time, there is variability in the correlations between CSF and plasma p-tau181 measured using different assays. Therefore, it is important to compare the p-tau181 results from the present study to previous work where p-

tau181 was quantified using the same assay, while comparisons to other p-tau forms or other p-tau181 assays are irrelevant. For example, a recent report (Ashton et al., *Alzheimers Dement* 2022) have demonstrated clearly lower correlations between CSF and plasma p-tau181 ($R=0.52$), measured using the same Quanterix Simoa assay.

Response: Thank you for your constructive comment. We agree that including individuals with MCI in the discussion section is inappropriate due to the heterogeneity, therefore we have removed such examples from the revised manuscript. After the literature review, we identified several studies that also demonstrate a non-significant difference in plasma $A\beta_{42}/A\beta_{40}$ between CU participants without AD and those with preclinical AD (Chong et al¹⁶ [Figure 1d]; Shen et al¹⁷ [Supplementary Table 9]) and updated the relevant reference with these articles (Page 10, Paragraph 3). For better illustration, the relevant data from the references are as follows:

Figure 1. Figure 1d from the original article by Chong et al, NCI $A\beta^-$ vs. NCI $A\beta^+$; NCI: non-cognitively impaired

Supplementary Table 9. General characteristics of cognitively normal controls from the SAS cohort

	AD non-converters	AD converters	P value
N	89	29	-
Age, y	69 ± 6	77 ± 6	< 0.001
Men (%)	49	41	0.452
APOE ε4 (%)	15	17	0.733
Education, y	12 ± 3	10 ± 4	0.001
Baseline MMSE score	29 ± 1	27 ± 2	< 0.001
Baseline plasma markers, pg/ml			
Aβ ₁₋₄₂	5.66 ± 1.68	7.14 ± 1.62	< 0.001
Aβ ₁₋₄₀	89.10 ± 28.66	117.80 ± 35.54	< 0.001
Aβ ₁₋₄₂ /Aβ ₁₋₄₀	0.066 ± 0.017	0.062 ± 0.010	0.408
P-tau181	2.10 ± 1.33	2.36 ± 1.41	0.748
NfL	16.46 ± 6.95	32.94 ± 20.96	< 0.001
GFAP	100.17 ± 48.95	171.10 ± 85.39	< 0.001

Abbreviations: SAS, Shanghai Aging Study; AD non-converters, individuals who did not convert to AD dementia during the follow-up period; AD converters, individuals who converted to AD dementia during the follow-up period.

Figure 2. Supplementary Table 9 from the original article by Shen et al

For the rationale behind the non-significant difference in plasma Aβ₄₂/Aβ₄₀ at baseline between the controls and pre-AD, we speculated several potential explanations. First, the discrepancies may be due to differences in inclusion criteria (CSF Aβ, PET Aβ, or clinical diagnosis) to determine preclinical AD across the studies. The preclinical AD population in the studies, which exhibited positive results for Aβ₄₂/Aβ₄₀ by Palmqvist et al. (JAMA Neuro, 2019)¹⁸, Schindler et al. (Neurology, 2019)¹⁹, and Palmqvist et al. (Alzheimers Dement 2023)²⁰, were biomarker-defined (i.e., dichotomous classification according to CSF or PET Aβ). However, our preclinical AD population was determined retrospectively based on the diagnosis made 8–10 years later, and patients' Aβ statuses at baseline were unknown. Since CSF and PET Aβ status contribute to Aβ status in the plasma, a significant difference in plasma Aβ₄₂/Aβ₄₀ between biomarker-defined

preclinical AD populations and biomarker-defined controls is more likely to be observed. Consistent with our study, Shen et al¹⁷ identified a non-significant difference in plasma A β 42/A β 40 at baseline between normal controls and AD converters (Figure 2) who were determined by AD diagnosis at follow-up. Additionally, another study demonstrated a significant difference in plasma A β 42/A β 40 levels between CSF-determined A β +CU and A β -CU, but the difference did not persist when the biomarker used to recruit participants was changed to PET A β ⁴, further emphasizing their role.

Second, the follow-up time to diagnosis was 8–10 years in our study, whereas other studies were either cross-sectional¹⁸ (thus, the time to diagnosis was undetermined) or mostly conducted over 4–6 years^{19,20}. Considering that plasma A β 42/A β 40 exhibits a small fold change between A β + and A β - individuals (a 10–15% reduction in plasma A β 42/A β 40 for A β + individuals compared with A β - individuals is observed, as opposed to the 50% reduction observed in CSF)²¹, the change in plasma A β 42/A β 40 over such a long period before symptom onset may not be strong enough for detection in our study. Furthermore, a model²² based on data from the BioFINDER cohort suggested that in plasma, A β 42 changes first, followed by changes in P-tau and the A β 42/A β 40 ratio before PET A β positivity. We hypothesized that our baseline data are in the intermediate state where plasma A β 42 has started to change but the A β 42/A β 40 ratio has not.

Collectively, the aforementioned factors may contribute to the non-significant difference in plasma A β 42/A β 40 at baseline between the controls and pre-AD individuals. However, the hypothetical speculations require further research. This result should be interpreted with caution as it is difficult to determine how much of the variability is controllable (e.g., through pre-analytical factors) and how much is due to inherent biological variability.

We apologize for any confusion regarding the relationship between the strict inclusion criteria of AD and the higher correlation of A β 42 in plasma and CSF. Unlike other studies that often employ a single A β status to include participants, our study used two biomarker criteria (CSF A β 42 and P-tau181/A β 42) to ensure a more precise selection of individuals with AD, which means that both the CSF A β 42 and P-tau181 status of participants were considered, making the participants in our study different from those in others. Differences in participants may contribute to the alteration in the correlation. Moreover, it has been well-documented that CSF A β 42 exhibits a relatively large dynamic range, whereas plasma A β 42 has a smaller dynamic range^{21,22}. Specifically, in our study, the biomarker inclusion criteria were set as CSF P-tau181/A β 42 > 0.14 and CSF A β 42 value < 500 pg/ml. The criteria may effectively narrow the range of CSF A β 42, whereas the range of plasma A β 42 may be less affected or unaffected. As a result, closer alignment of A β 42 values between two compartments was achieved, thereby enhancing the correlation of A β 42 between CSF and plasma. In contrast, other studies that used a dichotomous classification that was based on a single biomarker criterion may not display this effect. Additionally, factors such as racial disparity and preanalytical variation may contribute to this phenomenon. Notably, the result should be interpreted with caution because these explanations are purely hypothetical, and further research is required to elucidate the underlying mechanisms.

For the plasma and CSF correlation of P-tau181, as suggested by the reviewer, we conducted a literature review to identify comparable studies that examined the correlation using the Quanterix Simoa method and identified only three relevant articles. The reported correlation coefficients (r values) in these studies were 0.30²³, 0.52²⁴, and 0.65²⁵, respectively, with the latter being close to our findings (r=0.73).

Owing to the limited number of available studies, we extended our search to include studies that employed the prototype version of the Quanterix Simoa P-tau181 assay, known as P-tau181UGot, that was developed at the University of Gothenburg²⁴. Quanterix Simoa P-tau181 and P-tau181UGot utilize the same antibody pairs and platform, and they exhibit a nearly perfect correlation with each other^{24,26}. The correlation coefficients (r values) for P-tau181 between CSF and plasma, using the P-tau181UGot assay, ranged from 0.29 to 0.79^{12,24,27–29}. Notably, in the discovery cohort of Karikari's study²⁹, the correlation coefficients between blood and CSF P-tau181 were 0.7055 (plasma vs. CSF) and 0.7937 (serum vs. CSF), respectively, which were similar to or even higher than our findings (r = 0.73).

Furthermore, we identified that the correlation between CSF and plasma P-tau181 varied even within the same study. For instance, in one study³⁰, the reported correlation coefficient was 0.68 in the discovery cohort, while it dropped to 0.21 in the validation cohort, which employed the same assay as the discovery cohort. The underlying reasons for this inconsistency remain unclear and may be attributed to methodological variations, heterogeneity in the selected cohorts, or the utilization of specific inclusion criteria (e.g., CSF biomarkers vs. PET).

3. The conclusion in the last sentence of the abstract is not supported by the results. Since CSF data were not available at baseline and the accuracy of CSF biomarkers to predict AD dementia was not assessed, it is impossible to determine whether plasma biomarkers could substitute CSF biomarkers.

Response: Thank you for pointing this out. We have deleted this expression and changed the sentence to: “This study suggests that plasma A β 42, P-tau181, and NfL may be useful for predicting AD 8 to 10 years before clinical onset” in the revised manuscript (Page 2, Paragraph 1).

4. How often did study participant undergo clinical assessments? Could the authors rule out that individuals in the preAD groups progressed to AD dementia prior to 8-10y followup?

Response: Thank you for your valuable comment. In this longitudinal cohort, clinical assessments were performed every two years. To establish a biomarker model for predicting AD over an 8-10 year period, we used AD diagnosis at the 8th -10th years as the outcome; namely, we only chose individuals who developed AD dementia at the 8th -10th years as the case group, while excluding those who progressed to AD dementia before the 8th -10th years.

5. Presentation of statistical results could be further improved.

(a) Please reports odds ratio, 95% CI, AIC values, and p- values for models including individual biomarkers or their combination as well as AUC (95% CI) of individual biomarkers

(b) Were AUCs of individual biomarker significantly different AUC of the combined model

(c) The authors should consider including all ROC curves for the same outcome in one plot

(d) In the scatterplots, please highlight in different color the control and preAD groups

Response: Thank you for your constructive comment. We have added the models of individual biomarkers and compared them with the combination model (Page 7, Paragraph 1, Page 8, Paragraph 1, and Supplementary Table 2). Odds ratio, 95% CI, AIC values, and p values related to these models have been added (Supplementary Table 2). We replotted the graphs to include all ROC curves for the same outcome in one plot (Figure 5) and highlight control and case groups in different colors in scatterplots, as suggested (Figures 2 and 4).

References:

1. Rissin, D. M. *et al.* Single-molecule enzyme-linked immunosorbent assay detects serum proteins at subfemtomolar concentrations. *Nat. Biotechnol.* **28**, 595–599 (2010).
2. Bayoumy, S. *et al.* Clinical and analytical comparison of six Simoa assays for plasma P-tau isoforms P-tau181, P-tau217, and P-tau231. *Alzheimers Res. Ther.* **13**, 198 (2021).
3. Simrén, J. *et al.* The diagnostic and prognostic capabilities of plasma biomarkers in Alzheimer's disease. *Alzheimers Dement.* **17**, 1145–1156 (2021).
4. Verberk, I. M. W. *et al.* Plasma Amyloid as Prescreener for the Earliest Alzheimer Pathological Changes. *Ann. Neurol.* **84**, 648–658 (2018).
5. Vergallo, A. *et al.* Plasma amyloid β 40/42 ratio predicts cerebral amyloidosis in cognitively normal individuals at risk for Alzheimer's disease. *Alzheimers Dement. J. Alzheimers Assoc.* **15**, 764–775 (2019).
6. Wu, X. *et al.* Development of a plasma biomarker diagnostic model incorporating ultrasensitive digital immunoassay as a screening strategy for Alzheimer disease in a Chinese population. *Clin. Chem.* **67**, 1628–1639 (2021).
7. Quanterix Granted Breakthrough Device Designation from U.S. FDA for Blood-Based pTau-181 Assay for Alzheimer's Disease. *Quanterix* <https://www.quanterix.com/press-releases/quanterix-granted-breakthrough-device-designation-from-u-s-fda-for-blood-based-ptau-181-assay-for-alzheimers-disease/> (2023).
8. Lynch, A. P-Tau 181: Supporting the Development of LEQEMBI® for Alzheimer's Disease Therapy. *Quanterix* <https://www.quanterix.com/p-tau-181-supporting-the-development-of-leqembi-for-alzheimers-disease-therapy/> (2023).
9. Budd Haeberlein, S. *et al.* Two Randomized Phase 3 Studies of Aducanumab in Early Alzheimer's Disease. *J. Prev. Alzheimers Dis.* **9**, 197–210 (2022).

10. Mammel, A. *et al.* pTau181 plasma biomarker performance as an inclusion criterion in the RETHINK-ALZ and REFOCUS-ALZ trials in mild-to-moderate Alzheimer's disease.
11. Grønberg, M. *et al.* A Mass Spectrometry-based Proteomic Approach for Identification of Serine/Threonine-phosphorylated Proteins by Enrichment with Phospho-specific Antibodies. *Mol. Cell. Proteomics* **1**, 517–527 (2002).
12. Janelidze, S. *et al.* Head-to-head comparison of 10 plasma phospho-tau assays in prodromal Alzheimer's disease. *Brain* **146**, 1592–1601 (2023).
13. Janelidze, S. *et al.* Head-to-head comparison of 8 plasma amyloid- β 42/40 assays in Alzheimer disease. *JAMA Neurol.* **78**, 1375 (2021).
14. Chen, Z. *et al.* Learnings about the complexity of extracellular tau aid development of a blood-based screen for Alzheimer's disease. *Alzheimers Dement.* **15**, 487–496 (2019).
15. Hampel, H. *et al.* Blood-based biomarkers for Alzheimer's disease: Current state and future use in a transformed global healthcare landscape. *Neuron* S0896627323003902 (2023) doi:10.1016/j.neuron.2023.05.017.
16. Chong, J. R. *et al.* Plasma P-tau181 to A β 42 ratio is associated with brain amyloid burden and hippocampal atrophy in an Asian cohort of Alzheimer's disease patients with concomitant cerebrovascular disease. *Alzheimers Dement.* **17**, 1649–1662 (2021).
17. Shen, X.-N. *et al.* Plasma Glial Fibrillary Acidic Protein in the Alzheimer Disease Continuum: Relationship to Other Biomarkers, Differential Diagnosis, and Prediction of Clinical Progression. *Clin. Chem.* **69**, 411–421 (2023).
18. Palmqvist, S. *et al.* Performance of fully automated plasma assays as screening tests for Alzheimer disease-related β -amyloid status. *JAMA Neurol.* **76**, 1060 (2019).

19. Schindler, S. E. *et al.* High-precision plasma β -amyloid 42/40 predicts current and future brain amyloidosis. *Neurology* **93**, e1647–e1659 (2019).
20. Palmqvist, S. *et al.* An accurate fully automated panel of plasma biomarkers for Alzheimer's disease. *Alzheimers Dement.* **19**, 1204–1215 (2023).
21. Zetterberg, H. Biofluid-based biomarkers for Alzheimer's disease-related pathologies: An update and synthesis of the literature. *Alzheimers Dement.* **18**, 1687–1693 (2022).
22. Palmqvist, S. *et al.* Cerebrospinal fluid and plasma biomarker trajectories with increasing amyloid deposition in Alzheimer's disease. *EMBO Mol. Med.* **11**, e11170 (2019).
23. Tropea, T. F. *et al.* Plasma phosphorylated tau181 predicts cognitive and functional decline. *Ann. Clin. Transl. Neurol.* **10**, 18–31 (2023).
24. Ashton, N. J. *et al.* Plasma and CSF biomarkers in a memory clinic: Head-to-head comparison of phosphorylated tau immunoassays. *Alzheimers Dement.* alz.12841 (2022) doi:10.1002/alz.12841.
25. Álvarez-Sánchez, L. *et al.* Assessment of plasma and cerebrospinal fluid biomarkers in different stages of Alzheimer's disease and frontotemporal dementia. *Int. J. Mol. Sci.* **24**, 1226 (2023).
26. Chatterjee, P. *et al.* Diagnostic and prognostic plasma biomarkers for preclinical Alzheimer's disease. *Alzheimers Dement.* **18**, 1141–1154 (2022).
27. Milà-Alomà, M. *et al.* Plasma p-tau231 and p-tau217 as state markers of amyloid- β pathology in preclinical Alzheimer's disease. *Nat. Med.* **28**, 1797–1801 (2022).
28. Karikari, T. K. *et al.* Diagnostic performance and prediction of clinical progression of plasma phospho-tau181 in the Alzheimer's Disease Neuroimaging Initiative. *Mol. Psychiatry* **26**, 429–442 (2021).

29. Karikari, T. K. *et al.* Blood phosphorylated tau 181 as a biomarker for Alzheimer's disease: a diagnostic performance and prediction modelling study using data from four prospective cohorts. *Lancet Neurol.* **19**, 422–433 (2020).
30. Barthélemy, N. R., Horie, K., Sato, C. & Bateman, R. J. Blood plasma phosphorylated-tau isoforms track CNS change in Alzheimer's disease. *J. Exp. Med.* **217**, e20200861 (2020).

Reviewer #3 (Remarks to the Author):

I still do not believe this paper will have significant impact in the field or would be of particular interest to the readership. The Quanterix Simoa P-tau181 kit was approved by FDA in 2021 when first papers on plasma p-tau217 had just emerged and not many p-tau181 assays were available. We now have a much better understanding of how the performances of different plasma p-tau assays compare. Head-to-head studies assessing plasma p-tau assays have indicated that plasma p-tau217 measures (either MS-based or immunoassay-based) are better at detecting AD pathology and predicting future development of AD dementia than p-tau181 measures (reviewed in Hansson et al., *Nature aging*, 2023). Furthermore, the Quanterix Simoa p-tau181 is not even one of the best performing p-tau181 immunoassays and the same is true for the Quanterix Simoa A β immunoassay. While there is still a debate whether MS-based methods or immunoassays should be implemented for p-tau quantification, in my opinion the current Quanterix p-tau and A β immunoassay are very unlikely to be considered given that several better performing immunoassays are currently available.

REVIEWER COMMENTS

Reviewer #3 (Remarks to the Author):

I still do not believe this paper will have significant impact in the field or would be of particular interest to the readership. The Quanterix Simoa P-tau181 kit was approved by FDA in 2021 when first papers on plasma p-tau217 had just emerged and not many p-tau181 assays were available. We now have a much better understanding of how the performances of different plasma p-tau assays compare. Head-to-head studies assessing plasma p-tau assays have indicated that plasma p-tau217 measures (either MS-based or immunoassay-based) are better at detecting AD pathology and predicting future development of AD dementia than p-tau181 measures (reviewed in Hansson et al., Nature aging, 2023). Furthermore, the Quanterix Simoa p-tau181 is not even one of the best performing p-tau181 immunoassays and the same is true for the Quanterix Simoa A β immunoassay. While there is still a debate whether MS-based methods or immunoassays should be implemented for p-tau quantification, in my opinion the current Quanterix p-tau and A β immunoassay are very unlikely to be considered given that several better performing immunoassays are currently available.

Response: Thank you for your comment. We acknowledge your concerns, and we would like to provide some context for the significance and potential impact of our study.

This study leveraged a relatively accessible method to detect plasma A β 42, P-tau181, and NFL levels as reliable diagnostic and predictive markers for Alzheimer's disease (AD) in Asian populations. The P-tau181 assay has been approved by the FDA. This study provides evidence from a specific population that may not have been well-represented in previous studies to help extend plasma biomarkers of AD to clinical applications. Furthermore, it is noteworthy that a longer follow-up period was used in this study to assess the potential of plasma biomarkers in

predicting AD development during the preclinical phase. Therefore, we believe this work contributes to a broader understanding of AD biomarkers and their applicability in diverse populations.